# Impact of the COVID-19 pandemic on domiciliary care workers in Wales, UK: a data linkage cohort study using the SAIL Databank

Rebecca Cannings-John [1], Simon Schoenbuchner,[1] Hywel Jones,[2] Fiona V Lugg-Widger [1], Ashley Akbari [3], Lucy Brookes-Howell,[1] Kerenza Hood,[1] Ann John [4,5] Daniel Rh Thomas [6,7] Hayley Prout,[1] Michael Robling [1]

For numbered affiliations see end of article.

**Correspondence to**
Dr Rebecca Cannings-John; canningsrl@cardiff.ac.uk

## ABSTRACT

**Objectives** To quantify population health risks for domiciliary care workers (DCWs) in Wales, UK, working during the COVID-19 pandemic.

**Design** A population-level retrospective study linking occupational registration data to anonymised electronic health records maintained by the Secure Anonymised Information Linkage Databank in a privacy-protecting trusted research environment.

**Setting** Registered DCW population in Wales.

**Participants** Records for all linked DCWs from 1 March 2020 to 30 November 2021.

**Primary and secondary outcome measures** Our primary outcome was confirmed COVID-19 infection; secondary outcomes included contacts for suspected COVID-19, mental health including self-harm, fit notes, respiratory infections not necessarily recorded as COVID-19, deaths involving COVID-19 and all-cause mortality.

**Results** Confirmed and suspected COVID-19 infection rates increased over the study period to 24% by 30 November 2021. Confirmed COVID-19 varied by sex (males: 19% vs females: 24%) and age (>55 years: 19% vs <35 years: 26%) and were higher for care workers employed by local authority social services departments compared with the private sector (27% and 23%, respectively). 34% of DCWs required support for a mental health condition, with mental health-related prescribing increasing in frequency when compared with the prepandemic period. Events for self-harm increased from 0.2% to 0.4% over the study period as did the issuing of fit notes. There was no evidence to suggest a miscoding of COVID-19 infection with non-COVID-19 respiratory conditions. COVID-19-related and all-cause mortality were no greater than for the general population aged 15–64 years in Wales (0.1% and 0.034%, respectively). A comparable DCW workforce in Scotland and England would result in a comparable rate of COVID-19 infection, while the younger workforce in Northern Ireland may result in a greater infection rate.

**Conclusions** While initial concerns about excess mortality are alleviated, the substantial pre-existing and increased mental health burden for DCWs will require investment to provide long-term support to the sector's workforce.

## STRENGTHS AND LIMITATIONS OF THIS STUDY

⇒ Linking registration data to electronic health records in Wales provided a unique opportunity to explore health outcomes for domiciliary care workers (DCWs) working during the pandemic.
⇒ No DCWs dissented for their registration data to be made available for linkage, allowing a comprehensive population-level analysis.
⇒ Demographic data available by linked registration and healthcare datasets allowed assessment by key DCW subgroups.
⇒ Our study provides a broad range of health outcome estimates for DCWs over an extended period of the COVID-19 pandemic.
⇒ The recording of some data could have introduced bias, for example, coding for COVID-19 biased towards individuals presenting in primary or secondary healthcare.

## INTRODUCTION

During 2018–20, >950 000 people in the UK received domiciliary care from nearly 823 000 domiciliary care workers (DCWs) from over 10 100 home care providers and the demand is predicted to increase significantly over the next 20 years.[1] DCWs provide support to adults in their own home with household tasks, personal care or any activities that allow the individual to maintain independence. Support can demand close contact, assisting with bodily functions, managing continence, assisting with oral and dental care and providing other personal care. In the UK, domiciliary care can be funded publicly by local councils, privately funded or a combination of both.

These close working conditions with clinically vulnerable people, and the varied functions that DCWs perform in a non-institutional setting in a workforce with a

large number of employers, may increase exposure to COVID-19. A cross-sectional survey led by Ulster University found that during the early period of the COVID-19 pandemic (between May–July 2020 and November 2020–January 2021), both well-being and quality of working life deteriorated over all occupations including social care.[2] This decrease was again observed when care workers were surveyed between May and July 2021, with personal and work-related burnout reported to increase.[2] However, the survey did not aim to capture objectively recorded health outcomes specifically in DCWs, for example, positive rates of COVID-19 infection and the risk factors which contribute to these. In 2020, the Office for National Statistics (ONS) reported elevated mortality rates due to COVID-19 for 'care workers and home carers' compared with the general population in England and Wales.[3] In June 2020, Public Health England (PHE) reported COVID-19 infection rates for DCWs in line with general population rates.[4] Methodological differences between the two studies may explain contrasting estimates and neither study offered population coverage for a well-defined cohort of DCWs. Therefore, despite a policy interest in understanding level of risk for a population of workers potentially at increased likelihood of exposure to COVID-19 (and consequently the risk they could pose to clients), there was uncertainty about rates of COVID-19 infection or any other health outcome.

From 1 April 2020, it became mandatory for DCWs in the active Welsh workforce to be registered with Social Care Wales.[5] The population of working DCWs in Wales during the COVID-19 pandemic is therefore known, and can be linked to anonymised data, including health and administrative data held in the Secure Anonymised Information Linkage (SAIL) Databank (Swansea University), providing a unique opportunity to explore a range of health outcomes (including COVID-19 infection and mortality).

The Outcomes for Social Carers: an Analysis using Routine data (OSCAR) study aimed to use the registration data collected by Social Care Wales, individually linked to secure anonymised electronic health records (EHRs) to quantify population health risks for DCWs in Wales working during the COVID-19 pandemic. Specifically, we aimed to quantify rates of confirmed and suspected COVID-19 infection and, to fully establish the range of adverse health outcomes potentially affecting DCWs such as mental health including self-harm, issuing of fit notes, non-COVID-19 hospital admissions for respiratory conditions and COVID-19-related and all-cause mortality, as a consequence of the pandemic. We compared mental health outcomes (General practitioner contacts and prescriptions, and hospital admissions) within the pandemic period with the prepandemic situation to assess the effect of the pandemic. Similarly, we examined admissions over time for respiratory infection in the early pandemic to identify the potential miscoding of COVID-19 infection. Outcome variation was explored by demographics, work-related factors, lifestyle and comorbidities. To develop timely public health policy messages that could be extended to DCWs in other nations of the UK, using up-to-date DCW workforce data from Scotland, Northern Ireland (NI) and England, we compared these populations with the Welsh DCW workforce and assessed the generalisability of prevalence findings to each nation.

## METHODS

### Study design and data sources

This study is a population-level mixed-methods retrospective routine data linkage study with analysis guided by exploratory qualitative interviews. Qualitative findings are reported separately.[6] Registration data for all DCWs in Wales registered by 1 April 2020 are held by Social Care Wales (Domiciliary Social Care Worker (DSCW) data) and available via the SAIL Databank, a privacy-protecting trusted research environment (TRE), which uses a standardised split-file secure, encrypted anonymisation process.[7] These data were combined with data sources within the SAIL Databank (online supplemental material 1 for data sources used).

### Population

The study population was all registered DCWs resident in Wales on 1 March 2020 who did not subsequently opt-out to their data being transferred to the SAIL Databank and linked for research, either when DSCW data were added to the SAIL Databank or via their general practice (GP). Any DCWs not linked into the SAIL Databank with sufficient confidence were excluded. Additionally, individuals included in the dataset but registered as domiciliary care managers, adult care home managers or residential child care workers were excluded.

### Primary and secondary outcomes

The primary outcome was confirmed COVID-19 infection, defined as the earliest of the following events: a positive PCR test (Pathology Test Results Dataset (PATD)),[8] hospital admission (Patient Episode Database for Wales (PEDW)),[9] death registration from COVID-19 (main or secondary diagnostic International Classification of Diseases 10th Revision code=U07.1) (Annual District Death Extract (ADDE) from the ONS mortality register),[10] or a COVID-19 diagnosis (Welsh Longitudinal General Practice (WLGP)).[11]

Secondary outcomes were:
► Suspected COVID-19 infection (data sources: PATD, WLGP, PEDW and ADDE).
► Contacts for mental health and diagnoses, psychotropic medication and admissions (including self-harm) (data sources: WLGP and PEDW).
► Fit notes as a general marker of medically confirmed illness (data source: WLGP).
► Non-COVID-19 hospital admissions for respiratory conditions (data source: PEDW).

► Deaths involving COVID-19 and all-cause mortality (data source: ADDE).

In brief, we examined depression and anxiety symptoms and diagnoses,[12 13] and severe mental health in GP data and hospital admissions data.[14] Self-harm used GP data for definite and undetermined intent.[15 16] Fit notes also used GP data to cover medical/sickness/self-certificates issued to patients by general practitioners and included issues for returning to work. Hospital admissions for respiratory conditions were examined for potential miscoding of COVID-19 infection in the early pandemic (March–May 2020) and included any lower respiratory infection, pneumonia and influenza-like illness,[17] exacerbations of chronic obstructive pulmonary disease[18] and asthma.[19]

### Study period and follow-up

DCWs entered the cohort on 1 March 2020 and were followed up until 30 November 2021. We examined outcomes by waves of the pandemic; short-term outcomes from 1 March 2020 to 31 August 2020 (wave 1: reflecting a period before the relaxing of lockdown in Wales) and from 1 September 2020 to 28 February 2021 (wave 2). Long-term outcomes were examined to 30 November 2021. DCWs were followed up until the earliest of occurrence of death, first migration out of Wales, the outcome of interest or the end of the follow-up.

### DCW characteristics

We chose age, sex, ethnicity, health board, deprivation quintile and rurality as relevant characteristics for the DCWs using a number of data sources (online supplemental material 1). We also identified DCWs living with disability, any comorbidities,[14] shielding status and lifestyle factors such as body mass index (BMI) and smoking status. From the DSCW data, we were able to describe how individuals registered as a DCW (induction, registered with a qualification, competence confirmed by their manager), which reflects the different pathways open to care workers when mandatory registration was introduced.[20] Some pathways were transitional and are now no longer available (eg, registration following induction). We also identified whether a DCW lived with another DCW and their employment sector at the time of registration (private, third sector, local authority/ social services, recruitment/employment agency/ other). Several items from the DSCW were not available due to confidentiality considerations, such as length of time in the job, type of employment (eg, in social care, self-employed) and type of qualifications held (Qualifications and Credit Framework, National Vocational Qualification, etc).

A full coding list for outcomes is in online supplemental material 2 and also on Open Science Framework (OSF) (https://osf.io/9w6pe/) alongside codes for the comorbidities.

### Analysis

The linkage of DCWs EHR were described and the study population characterised using frequencies and percentages. The presence of key health outcomes were described by the short-term and long-term periods; prevalence estimates presented alongside 95% confidence intervals (CIs). This study is primarily designed to describe variation in outcomes in the DCW population, by key subgroups, rather than for prediction or causal models. Using multilevel Cox regression to model time to first confirmed COVID-19 infection, we estimated hazard ratios (HRs) with 95% CIs, to quantify the risk of confirmed COVID-19 by comorbidity status adjusting for age, sex and deprivation, and also by their employing sector adjusting for health board, rurality and qualification type. DCWs were censored at the time of the event, migration, death or at the study end.

To explore changes over time and the effect of the COVID-19 pandemic period, an interrupted time series approach was used to test the hypothesis that hospital admissions for non-COVID-19 respiratory infections would increase in comparison to the rates seen pre-March 2020, if there was misclassification of infections. Monthly data were divided into prepandemic (1 March 2016 to 28 February 2020) and the pandemic period (1 March 2020 to 31 October 2022). The changes in levels (baseline value of the outcome at time zero) and trends (the rate of change) of the prepandemic segment and pandemic segment were analysed using the *itsa* command in Stata. A Cumby-Huizinga test was used to determine the autocorrelation and was adjusted using Newey-West AC estimators. Using the same approach and as a post hoc analysis, the number of mental health GP contacts and prescriptions, and hospital admissions were also examined.

Equivalent up-to-date DCW workforce data from Scotland, NI and England were obtained and compared with the Welsh DCW cohort allowing us to generalise the estimate of confirmed COVID-19 cases in other nations.

To comply with disclosure control process and approvals, and protect the privacy of anonymised individuals, numbers <5 are suppressed. Percentages associated with small frequencies are rounded to the nearest 1%. Study findings were reported in accordance with applicable reporting guidelines for observational studies using administrative data (Strengthening the Reporting of Observational Studies in Epidemiology (STROBE) and REporting of studies Conducted using Observational Routinely-collected Data (RECORD)) (online supplemental material 3).[21 22] Data cleaning, cohort assembly and statistical analyses were performed using Structured Query Language (IBM Db2 V.11.1)[23] and R (V.4.1.0–V.4.1.3),[24 25] and Stata (V.17)[26] within the SAIL Databank privacy-protecting TRE.

### Patient and public involvement

Two stakeholder groups provided input to the project, an ongoing Study Advisory Group and an Implementation Reference Group. Both groups included membership

drawn from the domiciliary care sector and both contributed to interpretation of emerging results.

## RESULTS
### Study population
Data from 15 931 DCWs resident and working in Wales on 1 March 2020 were linked to EHR data (85% of them had GP records in SAIL). Records from 206 DCWs were excluded due to matching errors, leaving 15 725 (98.7%) DCWs included in the final analysis. The majority of DCWs were female (84%), of white ethnic background (95%), 43% had at least one comorbidity of which 22% had been recorded with mental health-related illness (table 1).

### Confirmed COVID-19 infection
Confirmed COVID-19 infection was 1.2% on 31 August 2020, rose to 14% by the end of February 2021 and reached 24% by the end of November 2021 (table 2). There was no evidence to suggest a miscoding of COVID-19 infection in the early pandemic, with no difference in admissions for non-COVID-19 respiratory conditions over time (online supplemental material 4). Confirmed COVID-19 infection was more prevalent in females, those who registered following the induction pathway compared with those with competency confirmed by a manager, those employed by the local authority social services at registration compared with those employed in the private sector (figure 1), those living with another DCW, those with a BMI >30 compared with those with a healthy BMI (18.5–24.9) and those living in an urban area (table 1). Confirmed COVID-19 infection was less prevalent in older DCW. Variation was also observed across health boards. After adjusting for age, sex and deprivation, COVID-19 infection was still more prevalent in those with a comorbidity (HR 1.08, 95% CI 1.01 to 1.15) and additionally adjusting for health board, rurality and qualification type, was more prevalent in those employed by the local authority social services at registration (HR 1.35, 95% CI 1.23 to 1.47).

### Suspected COVID-19 infection
Suspected COVID-19 infections increased over the pandemic from 2% in early pandemic to 15% by at the end of February 2021, rising to 24% by the end of November 2021 (table 2). Examining the prevalence of suspected COVID-19 infection added little to the clinical picture given the relatively widespread availability of COVID-19 testing across the follow-up period and uncertainty regarding the fields that were used (ie, just having a COVID-19 test was not a good indicator of having COVID-19 infection). No further modelling was therefore conducted.

### Mental health including self-harm
Contacts and diagnosis for mental health, psychotropic medication and admissions were 23% in August 2020, rising to 28% and 34% by the end of February and November 2021, respectively (table 2). Increases were also observed for self-harm (from 0.2% to 0.4%) over the same time period (table 2). The increase in mental health

issues over time was also apparent when examining the monthly number of mental health-related medication and contacts with general practitioners; prescribing mental health-related medication was recorded in 13% of all DCWs in March 2016 rising to 20% in March 2020, prescriptions increasing on average per month by 17.6 (95% CI 15.5 to 19.7, p<0.001) (figure 2A). No evidence of a change was observed immediately following the start of the pandemic (March to April 2020) but there was evidence of an increase in prescriptions after April 2020 (compared with the prepandemic) with an average monthly increase of 29.3 (95% CI 20.7 to 38.0, p<0.001). Similar patterns were observed for mental health-related GP contacts. Mental health-related hospital admissions were lower in March 2016 (around 15 admission per month) and significantly increased on average per month by 0.15 (95% CI 0.06 to 0.25, p=0.002) (figure 2B). There was evidence to suggest a decrease in admissions immediately following the start of the pandemic but increased on average by 0.99 admissions per month (95% CI 0.44 to 1.53, p<0.001), returning to prepandemic by the end of the study period.

Variations in mental health contacts were observed across many of the key demographics (eg, lower in males and older care workers), and consistently higher in care workers with a disability reported at registration, with a comorbidity or shielding (table 1). There was no variation in mental health contacts for the care workers by their qualification status; care workers in the local authority sector had slightly elevated risk of mental ill-health compared with those employed by the private sector at registration.

### Fit notes
The issuing of fit notes as a general marker of medically confirmed illness increased over the time period from 5% in August 2020 to 10% by at the end of February and 15% by the end of November 2021 (table 2). Fit notes were more prevalent in females, those who registered following the induction pathway or held a level 2 qualification (compared with those with competency confirmed by a manager), those employed by the local authority social services at registration (compared with those employed in the private sector), DCWs with a comorbidity or were shielding and those with a BMI >30 (compared with those with a healthy BMI 18.5–24.9) (online supplemental material 5). Unlike confirmed COVID-19 infection, fit notes were more likely in those that were older (aged 45 years and over). Both Betsi Cadwaladr and Powys health boards also had a higher rate of fit notes issued for DCWs, whereas Swansea Bay had less (when compared with Aneurin Bevan).

### Deaths
Both all-cause and COVID-19-related mortality involved 0.1% and 0.034%, respectively, of the care workers

**Table 1** Characteristics of DCWs with at least one confirmed COVID-19 infection and contacts for mental health and diagnoses, psychotropic medication and admissions in the COVID-19 pandemic between 1 March 2020 and 31 November 2021

| | DCWs N=15 725 | | Confirmed COVID-19* N=3698 | | Mental health* N=5300 | |
|---|---|---|---|---|---|---|
| | N | % | N | % | N | % |
| **DCW characteristics** | | | | | | |
| **Sex†** | | | | | | |
| Female | 13 253 | 84.3 | 3226 | 24.3 | 4685 | 35.4 |
| Male | 2472 | 15.7 | 472 | 19.1 | 615 | 24.9 |
| **Age (at registration)†** | | | | | | |
| Under 35 years | 4425 | 28.1 | 1164 | 26.3 | 1542 | 34.9 |
| 35 to <45 years | 2822 | 17.9 | 795 | 28.2 | 977 | 34.6 |
| 45 to <55 years | 3955 | 25.2 | 897 | 22.7 | 1398 | 35.4 |
| 55 years and over | 4523 | 28.8 | 842 | 18.6 | 1383 | 30.6 |
| **Ethnicity†** | | | | | | |
| White | 15 012 | 95.5 | 3537 | 23.6 | 5128 | 34.2 |
| Black | 142 | 0.9 | 35 | 24.7 | 22 | 15.5 |
| Mixed | 127 | 0.8 | *** | *** | *** | *** |
| Asian | 108 | 0.7 | 25 | 23.2 | 18 | 16.7 |
| Other | 66 | 0.4 | *** | *** | *** | *** |
| Mixed/Other | 193 | 1.2 | 42 | 21.8 | 46 | 23.8 |
| Not recorded | 270 | 1.7 | 59 | 21.9 | 86 | 31.9 |
| **Lives with DCW?†—Yes** | 975 | 6.2 | 254 | 26.1 | 312 | 32.0 |
| No | 14 750 | 93.8 | 3444 | 23.4 | 4988 | 33.8 |
| **Has disability?†—Yes** | 188 | 1.2 | 30 | 16.0 | 91 | 48.4 |
| No | 14 141 | 89.9 | 3310 | 23.4 | 4714 | 33.3 |
| Not recorded/Prefer not to answer | 1396 | 8.9 | 358 | 25.6 | 495 | 35.5 |
| **Qualifications as recorded at registration†** | | | | | | |
| Competence confirmed | 1982 | 12.6 | 438 | 22.1 | 646 | 32.6 |
| Induction framework | 2984 | 19.0 | 737 | 24.7 | 1094 | 36.7 |
| Qualification level 2 | 6935 | 44.1 | 1672 | 24.1 | 2328 | 33.6 |
| Qualification level 3 | 3340 | 21.2 | 752 | 22.5 | 1077 | 32.3 |
| Qualification level 4+ | 441 | 2.8 | 88 | 20.0 | 140 | 31.8 |
| Not recorded | 43 | 0.3 | 11 | 25.6 | 15 | 32.7 |
| **Employment sector as recorded at registration†** | | | | | | |
| Private | 6592 | 41.9 | 1480 | 22.5 | 2154 | 32.7 |
| Third sector | 4267 | 27.1 | 991 | 23.2 | 1454 | 34.1 |
| Local authority-social services | 3306 | 21.0 | 876 | 26.5 | 1159 | 35.1 |
| Recruitment/Employment agency | 148 | 0.9 | 43 | 25.0 | 65 | 37.8 |
| Health/Other | 24 | 0.2 | | | | |
| Not recorded | 1388 | 8.8 | 308 | 22.2 | 468 | 33.7 |
| **Smoking status‡** | | | | | | |
| Non-smoker | 6342 | 40.3 | 1638 | 25.8 | 2076 | 32.7 |
| Current smoker | 4374 | 27.8 | 875 | 20.0 | 1781 | 40.7 |
| Ex-smoker | 3509 | 22.3 | 1185 | 23.7 | 1443 | 28.8 |
| Not recorded | 1500 | 9.5 | | | | |
| **Body mass index‡** | | | | | | |
| Underweight (<18.5) | 256 | 1.6 | 48 | 18.8 | 99 | 38.7 |
| Healthy weight (18.5–24.9) | 3173 | 20.2 | 685 | 21.6 | 1045 | 32.9 |
| Overweight (25–29.9) | 3652 | 23.2 | 847 | 23.2 | 1341 | 36.7 |

Continued

**Table 1** Continued

| | DCWs N=15725 | | Confirmed COVID-19* N=3698 | | Mental health* N=5300 | |
|---|---|---|---|---|---|---|
| | N | % | N | % | N | % |
| Obese (≥30) | 5311 | 33.8 | 1398 | 26.3 | 2320 | 43.7 |
| Not recorded | 3333 | 21.2 | 720 | 21.6 | 495 | 14.9 |
| Any comorbidity?‡ § ¶—Yes | 6804 | 43.3 | 1652 | 24.3 | 3205 | 47.1 |
| No | 8921 | 56.7 | 2046 | 22.9 | 2095 | 23.5 |
| Mental health-related illness** | 3431 | 21.8 | – | – | – | – |
| Asthma | 2699 | 17.2 | – | – | – | – |
| CHD | 281 | 1.8 | – | – | – | – |
| CKD | 132 | 0.8 | – | – | – | – |
| COPD | 299 | 1.9 | – | – | – | – |
| Epilepsy | 238 | 1.5 | – | – | – | – |
| Osteoporotic fracture | 462 | 2.9 | – | – | – | – |
| Stroke | 135 | 0.9 | – | – | – | – |
| Diabetes | 824 | 5.2 | – | – | – | – |
| On COVID-19 shielding list?††—Yes | 635 | 4.0 | 129 | 20.3 | 267 | 42.1 |
| No | 15090 | 96.0 | 3569 | 23.7 | 5033 | 33.4 |
| Area-level characteristics‡‡ §§ | | | | | | |
| Resident health board | | | | | | |
| Aneurin Bevan | 2819 | 17.9 | 758 | 26.9 | 980 | 34.8 |
| Betsi Cadwaladr | 3683 | 23.4 | 665 | 18.1 | 1140 | 31.0 |
| Cardiff and Vale | 1796 | 11.4 | 475 | 26.5 | 650 | 36.2 |
| Cwm Taf Morgannwg | 2702 | 17.2 | 755 | 27.9 | 1078 | 39.9 |
| Hywel Dda | 1965 | 12.5 | 392 | 20.0 | 561 | 28.6 |
| Powys | 662 | 4.2 | 104 | 15.7 | 105 | 15.9 |
| Swansea Bay | 2098 | 13.3 | 549 | 26.2 | 786 | 37.5 |
| Deprivation quintile¶¶ | | | | | | |
| 1 (most deprived) | 4168 | 26.5 | 1143 | 27.4 | 1541 | 37.0 |
| 2 | 3970 | 25.2 | 991 | 25.0 | 1392 | 35.1 |
| 3 | 3301 | 21.0 | 697 | 21.1 | 1005 | 30.5 |
| 4 | 2554 | 16.2 | 503 | 19.7 | 780 | 30.5 |
| 5 (most affluent) | 1732 | 11.0 | 364 | 21.0 | 582 | 33.6 |
| Rurality classification | | | | | | |
| Rural | 4932 | 31.4 | 947 | 19.2 | 1435 | 29.1 |
| Urban | 10793 | 68.6 | 2751 | 25.5 | 3865 | 35.8 |

*Between 1 March 2020 and 31 November 2021.
†Data source=Domiciliary Social Care Worker data and include data that are self-reported by DCWs on 1 April. If missing, then other sources (ONS 2011 Census Wales, WLGP, PEDW, Emergency Department Dataset) were used to impute (at index date=1 March 2020).
‡Data source=WLGP.
§Data source=PEDW.
¶Most recent occurring between 1 January 2000 and 1 March 2020.
**Contacts and diagnoses for mental health and admissions.
††People at a higher risk of getting seriously ill from COVID-19.
‡‡Based on residential address on 1 March 2020.
§§Data source=Welsh Demographic Service Dataset.
¶¶The Welsh Index of Multiple Deprivation (WIMD) is the official measure of relative deprivation for small areas in Wales. Quintiles are derived by proportioning the derivation score into five equal groups.
***Numbers suppressed as <10. Mixed and other categories are combined.
CHD, coronary heart disease; CKD, coronary kidney disease; COPD, chronic obstructive pulmonary disease; DCW, domiciliary care worker; PEDW, Patient Episode Dataset for Wales; WLGP, Wales Longitudinal General Practice.

over the long-term study period (1 March 2020 and 30 November 2021) (table 2).

### Comparison with other UK nations

Up-to-date DCW workforce from Scotland[27] and England[28] show comparable case-mix of DCW populations to (and in some cases identical to) that presented for Wales with respect to age, sex and ethnicity (online supplemental material 6) and we would expect the prevalence of confirmed COVID-19 in these nations to therefore be similar. For DCWs in

**Table 2** Key outcomes by pandemic wave

| Outcome | Short term (wave 1) 1 March to 31 August 2020 | | Short term (waves 1 and 2) 1 March 2020 to 28 February 2021 | | Long term 1 March 2020 to 30 November 2021 | |
|---|---|---|---|---|---|---|
| | N | % | N | % | N | % |
| Confirmed COVID-19 infection | 182 | 1.2 | 2196 | 14.0 | 3698 | 23.5 |
| Suspected COVID-19 infection | 326 | 2.1 | 2335 | 14.8 | 3822 | 24.3 |
| Mental health* | 3641 | 23.2 | 4392 | 27.9 | 5300 | 33.7 |
| Self-harm | 20 | 0.1 | 35 | 0.2 | 64 | 0.4 |
| Fit notes | 857 | 5.4 | 1551 | 9.9 | 2297 | 14.6 |
| Non-COVID-19 respiratory infections† | 126 | 0.8 | 300 | 1.9 | 467 | 3.0 |
| All-cause mortality | ‡ | ‡ | ‡ | ‡ | 23 | 0.1 |

*Contacts and diagnosis for mental health, psychotropic medication and admissions.
†Respiratory infections not necessarily recorded as COVID-19 infection.
‡Short-term results suppressed as de-identifiable by subtraction from the long-term results.

England based on a population of 590 000, the impact of mental health illness is around 200 000. However, data from NI showed that DCWs are younger on average (mode 20–29 years),[29] and the prevalence of COVID-19 infection may have been greater in this DCW workforce.

## DISCUSSION
### Principal findings

Confirmed COVID-19 infection rates increased from 14% during the first two pandemic waves (1 March 2020 to 28 February 2021) to 24% by the end of November 2021. Infection rates varied by personal characteristics of care workers (lower in males and in older care workers), by where care workers lived (lower in health boards such as Powys, in rural areas, in more affluent areas) and were higher for care workers employed by local authority social services departments compared with staff employed

in the private sector. Suspected COVID-19 infections similarly increased to 24% over the study period. One-third (34%) of all care workers required support for a mental health condition, with rates of attending a GP or receiving a relevant prescription higher compared with the 4 years preceding pandemic onset. Rates in mental health contacts varied by key demographics and were consistently higher in care workers with a comorbidity, disability or who were shielding. Local authority care workers had a slightly elevated risk of mental health problems compared with those employed by the private sector. Events for self-harm also increased, from 0.2% to 0.4%. The issuing of fit notes as a general marker of medically confirmed illness increased by 10% over the COVID-19 pandemic and were more prevalent in female DCWs, those with an underlying comorbidity or at a higher risk of getting seriously ill from COVID-19. Unlike confirmed COVID-19 infection, fit notes were more likely in those

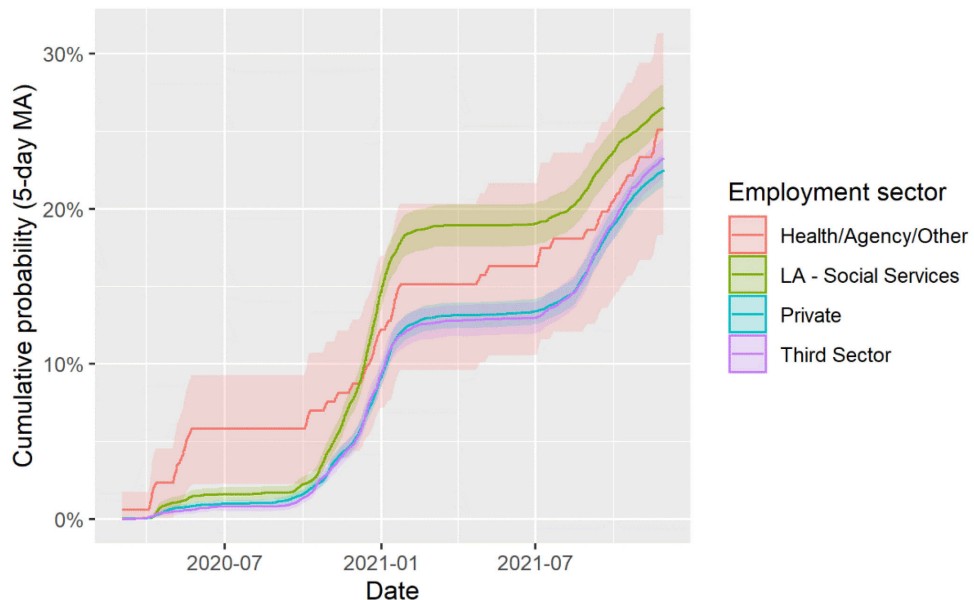

**Figure 1** Cumulative probabilities of confirmed COVID-19 infection over time by employment sector for domiciliary care workers. LA, local authority; MA, moving average.

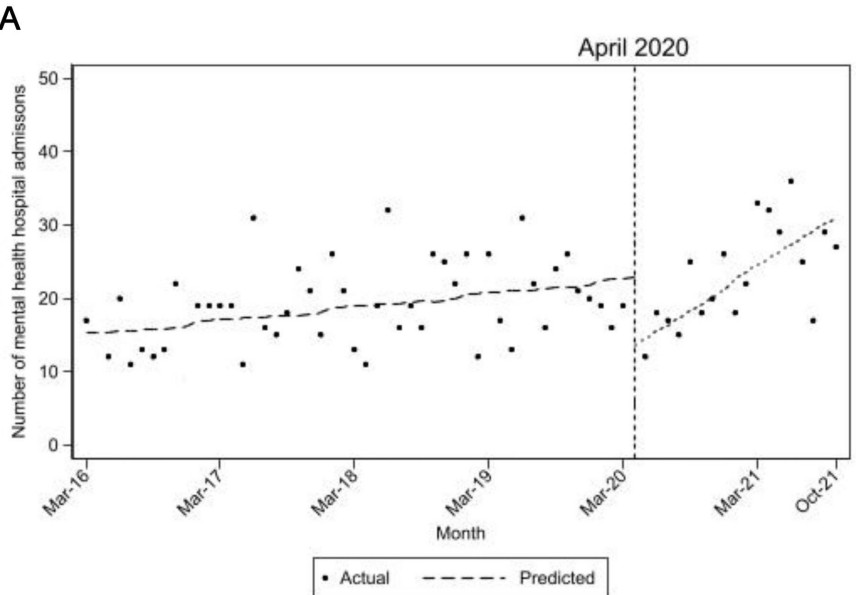

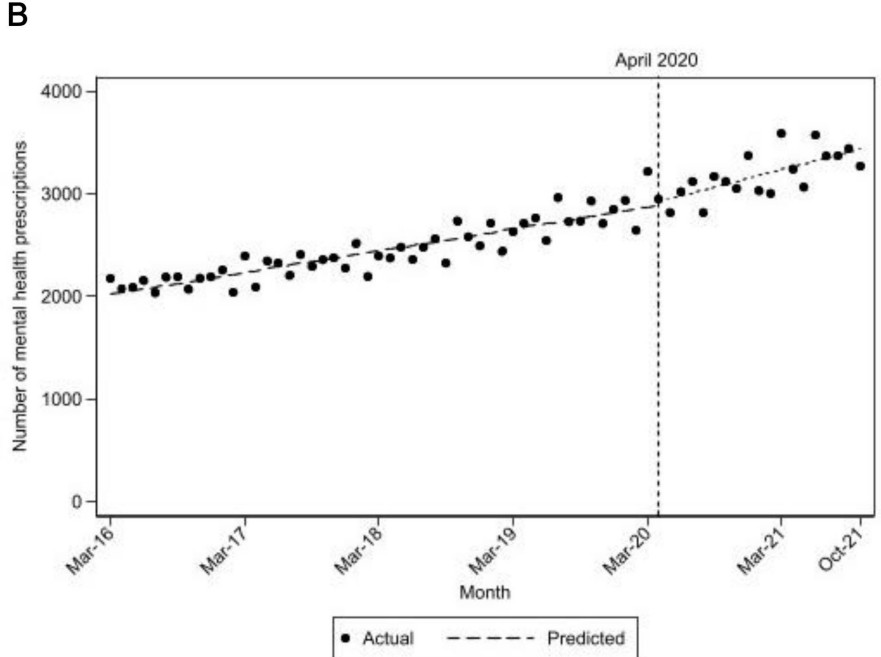

**Figure 2** Changes over time (March 2016–October 2021) in mental health-related hospital admissions (A) and GP prescriptions (B) for domiciliary care workers. GP, general practice.

that were older (aged 45 years and over). There was no evidence to suggest a miscoding of COVID-19 infection in the early pandemic. The mortality rate among care workers was no greater than that observed among the general population aged 15–64 years in Wales (which was 0.034%). Data from DCW workforces in England and Scotland showed a similar demographic with respect to age, sex and ethnicity. The impact of COVID-19 infection would therefore be similar to that observed in Wales. The NI workforce was younger and therefore we may expect a high prevalence rate.

### Strengths and weaknesses

A principal strength of the study was the comprehensive and contemporaneous coverage of the DCW population in Wales through linkage to registration data. This contrasts with approaches reliant on census data which may be incomplete, at risk of being out of date or on sample surveys with varying levels of non-response.[3 4] With few records lost to linkage failure or opt-out, the study cohort reflects well the population of DCWs reported by the regulator.[30] Access to EHRs in the SAIL Databank allowed for broader assessment of health than previously

reported by ONS, which understandably focused on mortality in the early period of the pandemic.[3] Accessing SAIL data used coding developed for existing conditions and novel indications (eg, COVID-19) enabling rapid application and greater confidence than coding developed from scratch.[31 31]

The registration process was introduced by Social Care Wales to meet regulatory rules that came into force from 1 April 2020.[5] Registration as a DCW reflected the role's focus on providing care whereas workers providing support only were not required to register. However, our qualitative interviews revealed greater heterogeneity in the registered workforce than anticipated.[6] Workers in supported shared living homes were registered, as well as workers travelling to care for individuals living in their own home, detail unavailable from registration data alone. In all cases, members of the study cohort are registered as DCWs and despite some potential differences in working role among the cohort, common factors are likely to involve close proximity with clients in their domestic setting. Differences may include provision of intimate personal care and assistance with bodily functions by DCWs only. Our inability to further differentiate between workers consistently working in either a care or support-only role during the study period is a limitation, especially if this represents a difference in risk exposure. Integrating qualitative interviews (which have been separately reported) with the quantitative routine data analyses has provided insight into the working experience and context of DCWs and how that may translate to variability in infection risk and mitigation.[6] It has illuminated the heterogeneity of roles being undertaken by those registered under the umbrella role of DCW and therefore we would consider a strength.

Administrative records do not usually capture subjective patient-reported outcomes or well-being concerns. This may be a greater consideration if care workers felt reluctant or less able to present concerns to healthcare professionals during the pandemic, as has been observed in other settings.[32] Both quantitative and qualitative studies, involving a range of community-based health and social care staff provide ample evidence of impacts of the pandemic on the well-being of professionals.[2 6 33–35] Findings from our routine data study build a more complete picture of the health consequences of the pandemic for care workers.

Outcome rates varied considerably based on characteristics and circumstances of care workers, such as age and geographical location. Such differences may not be directly attributable to the working conditions of care workers or even directly addressable. Nevertheless, it highlights that risks for care workers remain a vital consideration for employers.

Some methodological caveats include that we could not distinguish between DCWs who did not consult with their GP or where their GP had not contributed data to SAIL. This will result in downward biased GP outcome rates (eg, if a mental health diagnosis truly occurred then a mental health diagnosis will be misclassified as having not occurred). Second, if missing GP data are not at random (eg, if there is less GP coverage in some regions), it is more likely that we would have missed mental health diagnosis or fit notes from DCWs with certain characteristics, resulting in bias in our parameter estimates (ORs/HRs) for these outcomes. Third, DCWs were followed up for different durations (due to migration in and out of Wales or from contributing/non-contributing GPs, or deaths). While the time to event analyses (and corresponding HRs) account for this by censoring, summary statistics and ORs do not. Lastly, cohort ageing and differences in cohort entry or exit dates would likely lead to upward bias in the overall trends in the time series analyses.

## Findings in context

In contrast to our findings, ONS data from the initial 3-month period of the pandemic suggested that mortality rates for individuals employed as care workers and home carers were higher than those found in the general population.[3] However, data on occupational classification were missing for a large proportion dying with some COVID-19 involvement and, even where available, data on occupation may have been incorrect due to role and job changes since census data were provided (ie, every 10 years). Our findings are more in line with the initial estimates provided by a PHE survey of DCWs in July 2020, where the rate of confirmed COVID-19 on PCR testing (regardless of symptoms) was low (0.1%, 95% CI 0.01 to 0.36%).[4] The PHE study differed from out study in several ways. In the PHE study, DCWs were identified through purposive sampling of care providing organisations by region (ie, to achieve similar numbers of respondents by region), initially approached based on convenience. Within participating organisations, a convenience sample of staff were approached to take part in the survey which involved staff returning a nasal swab and short questionnaire during a 2-week study window (2–16 June 2020). DCW staff included in the PHE study were 84.3% female, with a median age of 41 years and 75.8% of those providing details about ethnicity were white. Two of the 2015 DCWs returning a swab in the 2-week study period were positive for SARS-CoV-2 on PCR testing, while 41 DCWs reported symptoms of COVID-19 in the 14 days prior to the swab being taken.

Differing study methodologies may reduce the value of direct comparisons, while also offering different levels of insight. Clarity of the study population is one such parameter. At the commencement of the pandemic consideration of work-place characteristics, opportunities to mitigate risk and rapidly emerging evidence of infection and associated mortality drove interest in social care and in our study, the work of DCWs. Subsequently, a semi-quantitative job exposure matrix (COVID-19-JEM) was developed to estimate the likelihood of workers becoming infected with SARS-CoV-2 in an occupational setting.[36] 'Home workers' and 'home carers' were assessed as scoring 14 on the matrix, indicating a relatively high risk

of exposure due to characteristics inherent in their job. However, the threshold used (13+) would include half of the UK workforce and the matrix's factors for transmission risk and mitigation could clearly mask a heterogeneity of risk as suggested in our own qualitative work.[6] An analysis of occupational differences in SARS-CoV-2 infection used the UK ONS COVID-19 infection survey (CIS) data for adults aged 20–64 years.[37 38] The CIS period covered (April 2020 to November 2021) was almost equivalent to that in our own study. Using the four-digit Standard Occupational Classification (SOC) 2010, the study reported an elevated risk for social care staff (adjusting for multiple demographic factors) with an HR of 1.14 (95% CI 1.04 to 1.24) when compared with non-essential workers. Of the 8005 social care staff included in the CIS, 690 (8.6%) had at least one positive PCR test during the study period. Interestingly, the elevated odds of an infection for social care staff reduced over the time assessed (as it did for healthcare workers), while it persisted for some other occupations such as education. Finally, in comparison, the report found equivocal evidence overall regarding infection rate for healthcare support workers (HR 1.13; 95% CI 0.96 to 1.32).

Comparing rates across studies may not always be helpful due to the differences in methods used and the time intervals covered. For example, while the ONS survey used RT-PCR testing, other studies have used other methods (eg, self-administered point-of-care lateral flow immunoassay testing in REal Time Assessment of Community Transmission-2 (REACT-2)).[38 39] Nevertheless, the higher rate of confirmed infection for DCWs in our study compared with social care staff in the ONS survey cannot be attributable to differences in time period covered as they were virtually equivalent. While DCWs in Wales were likely to have tested several times in that time period, so would have participants in the ONS study (the CIS involved monthly testing after an initial baseline set of 5 weekly assessments). The lack of granularity associated with SOC codes and COVID-19-JEM to reflect contamination risk in specific settings is probably a limiting factor in such analyses. Our study sample remained specifically focused on workers registered as DCWs, which may reflect a higher level of occupational risk than for other workers classed as in caring personal services or in social care. In this context, within study population differences (eg, by sex, gender) are less problematic to interpret, especially given the comprehensive coverage of the DCW population and the large numbers of well-characterised DCWs included.

Previous research on care worker health outcomes has often focused on musculoskeletal problems and dealing with challenging behaviour.[40] Nevertheless, prepandemic work has emphasised some broader harms experienced by a range of community-based home care workers (eg, infection control) and the benefits that workers derive from their work.[41–43] Sterling *et al* used US national Centers for Disease Control and Prevention Behavioral Risk Factor Surveillance Survey data to explore outcomes of home healthcare workers employed by home care agencies.[41] Despite role differences between the US sample and DCWs in Wales, the pre-pandemic levels of mental ill-health are interestingly similar. The reasons for such similarities are unclear and could simply reflect broader population similarities in the prevalence of mental ill-health rather than occupationally driven factors.

## CONCLUSION

This study presents evidence of the direct and indirect impact of the COVID-19 pandemic on DCWs in Wales. Higher rates of confirmed COVID-19 infection among DCWs did not translate into mortality rates greater than for the broader Welsh population. High baseline rates of mental ill-health further increased over time, a burden that fell unevenly across the workforce. It remains to be determined whether any facet of the employment role, such as staff training, occupational risk assessment or testing procedures, may have contributed to these differences or provided an opportunity to intervene. Evidence from our own work and that of others supports the value of co-produced solutions which draw on the direct experiences of care workers to support occupational related well-being.[44] Systemic drivers (eg, public funding for social care, staffing levels, levels of pay) and situational aspects of the role such as peripatetic working will not change quickly or even at all. With few evidence-based supportive approaches tailored to the circumstances of care workers[45 46] optimising or innovating approaches to support the UK community of care workers may offer considerable long-term benefits to workforce and clients alike.

**Author affiliations**
[1]Centre for Trials Research, Cardiff University, Cardiff, UK
[2]Division of Population Medicine, Cardiff University, Cardiff, UK
[3]Faculty of Medicine, Health & Life Science, Swansea University Medical School, Swansea, UK
[4]Health Data Research UK | Administrative Data Research Wales, Swansea University, Swansea, UK
[5]DECIPHer—Centre for Development, Evaluation, Complexity and Implementation in Public Health Improvement, Cardiff University, Cardiff, UK
[6]Communicable Disease Surveillance Centre, Public Health Wales, Cardiff, UK
[7]Cardiff Metropolitan University, Cardiff, UK

**Acknowledgements** The study team gratefully acknowledge the work of the Study Advisory Group for providing independent oversight and input; the Implementation Reference Group for reviewing early findings and co-producing recommendations for short-term and long-term outcomes; and all domiciliary care workers who contributed their data for inclusion in the study analyses. This study makes use of anonymised data held in the Secure Anonymised Information Linkage (SAIL) Databank. We would like to acknowledge all the data providers who make anonymised data available for research. We acknowledge the support of Social Care Wales in advising us at different stages of the study, supporting access to the registration data and aiding our reflection on study findings.

**Contributors** RC-J and MR were co-chief investigators of the study and led the drafting of the main manuscript. MR, RC-J, AJ, FVL-W, AA, KH, LB-H and DRhT conceived and contributed to the design of the study. AA was responsible for

managing access to data within the SAIL Databank and advising on constituent datasets. HJ was responsible for data management and preparing data for analysis. FVL-W was responsible for study management. SS and RC-J were responsible for developing the statistical analysis plan and conducting the main analysis. All authors contributed to the design and conduct of the study and interpretation of the analyses. All authors were responsible for reviewing and revising drafts of the manuscript and providing final approval. RC-J and MR are responsible for the overall content as the guarantors.

**Funding** This research was funded by the Economic & Social Research Council (ESRC), as part of UK Research & Innovation's rapid response to COVID-19 (ES/V015206/1). The Centre for Trials Research receives funding from Health and Care Research Wales and Cancer Research UK. The data acquisition and COVID-19 research was supported by Health Data Research UK (HDR-9006), which receives its funding from HDR UK, and the ADR Wales programme of work) funded ADR UK (grant ES/S007393/1).

**Competing interests** None declared.

**Patient and public involvement** Patients and/or the public were involved in the design, or conduct, or reporting, or dissemination plans of this research. Refer to the 'Methods' section for further details.

**Patient consent for publication** Not applicable.

**Ethics approval** The study received approval from the School of Medicine Research Ethics Committee, Cardiff University (SMREC 20/114]). Approval was obtained from the SAIL Information Governance Review Panel (IGRP), Swansea University (Project ID: 1126) and data access was granted to named staff. Approval from the ONS Research Accreditation Panel (ONS RAP) was provided (16 December 2020) and approval from Social Care Wales was provided (26 January 2021). This process was generic and not specific to the OSCAR study.

**Provenance and peer review** Not commissioned; externally peer reviewed.

**Data availability statement** Data may be obtained from a third party and are not publicly available. The routine data used in this study are available in the SAIL Databank at Swansea University, Swansea, UK. All proposals to use SAIL data are subject to review by an IGRP. Before any data can be accessed, approval must be given by the IGRP. The IGRP gives careful consideration to each project to ensure proper and appropriate use of SAIL data. When access has been approved, it is gained through a privacy protecting safe haven and remote access system referred to as the SAIL Gateway. SAIL has established an application process to be followed by anyone who would like to access data via SAIL: https://www.saildatabank.com/application-process. This study has been approved by the IGRP as project 1126.

**ORCID iDs**
Rebecca Cannings-John http://orcid.org/0000-0001-5235-6517
Fiona V Lugg-Widger http://orcid.org/0000-0003-0029-9703
Ashley Akbari http://orcid.org/0000-0003-0814-0801
Ann John http://orcid.org/0000-0002-5657-6995
Daniel Rh Thomas http://orcid.org/0000-0002-2426-5893
Michael Robling http://orcid.org/0000-0002-1004-036X

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
