## [Reviewer comments · BMJ Open]

ARTICLE DETAILS

TITLE (PROVISIONAL)	Impact of the COVID-19 pandemic on domiciliary care workers in Wales, UK: a data linkage cohort study using the SAIL Databank
AUTHORS	Cannings-John, Rebecca; Schoenbuchner, Simon; Jones, Hywel; Lugg-Widger, Fiona; Akbari, Ashley; Brookes-Howell, Lucy; Hood, Kerensa; John, Ann; Thomas, Daniel; Prout, Hayley; Robling, Michael

VERSION 1 – REVIEW

REVIEWER	Warters, Austin
REVIEW RETURNED	17-Feb-2023

GENERAL COMMENTS	Really important paper and valuable contribution to understanding the impact of Covid-19 on this specific workforce. Some minor comments Line 6 - is dom care provided publicly funded by the state, or do people make co-payments, or is some of it privately funded - might be useful to include if known Line 27 - could it be explained how roles increased?
---

REVIEWER	Hoedl, Manuela Medical University of Graz, Institute of Nursing Science
REVIEW RETURNED	21-Feb-2023

GENERAL COMMENTS	Thank you for giving me the possibility to review your manuscript regarding the “The impact of the COVID-19 pandemic on Domiciliary Care Workers in Wales (UK): a data linkage cohort study using the SAIL Databank”. This is a very interesting topic. In general, the paper is very well done and written. Nevertheless, there are three main aspects, from my point of view, that are missing in this manuscript. (1) Please go more into depth in the introduction, why it is important to investigate • your topic,• this special sample,• the different primary and secondary outcomes, as well as• the comparison with other UK nations or• the comparison with the pre pandemic situation. I miss a clear argumentation, why this is important. (2) Please stick to your aim including all variables, the order of the variables in the method, result as well as in the discussion section.
---

(3) Please discuss the following outcome variables (Especially the outcome variables) also in the discussion section.

- Suspected COVID-19 infection:
- Confirmed COVID-19 infection:
- Key health outcomes: fit notes
- Explore outcome variation...
- Examine generalizability to other UK nations

Abstract: Objectives

You wrote: To quantify population health risks for domiciliary care workers in Wales working during the COVID-19 pandemic.

In your aim at the end of the introduction section you wrote the following: The OSCAR study aimed ...to understand the impact of COVID-19 on DCWs in Wales.

So do you quantify something or do you want to understand the impact of the pandemic?

Abstract: Primary and secondary outcome measures

You wrote: Our primary outcome was confirmed COVID-19 infection; secondary outcomes included contacts for mental health, fit notes, respiratory infections not necessarily recorded as COVID-19, deaths involving COVID-19, all-cause mortality and suspected COVID-19.

In your method section you wrote the following: The primary outcome was confirmed COVID-19 infection, defined as the earliest of the following events: a positive polymerase chain reaction (PCR) test, hospital admission, death registration from COVID- 19, or a COVID-19 diagnosis Read code.

So what is with , hospital admission, death registration from COVID- 19, or a COVID-19 diagnosis Read code as primary outcome? And I cannot find the numbers for deaths involving COVID-19 in your result section. Sorry. You only wrote once: The mortality rate amongst care workers was no greater than that observed amongst the general population of 15- to 64-year-olds in Wales (which was 0.034%). But here it is not stated that it is deaths involving COVID-19.

STRENGTHS AND LIMITATIONS OF THIS STUDY

You wrote:

- Relatively little is known about the objective health of the UK's domiciliary care workforce (DCWs), either pre- or during the COVID-19 pandemic; our study aims to address this deficit. Interesting, as there is no information about the pre-pandemic situation except for Mental health illness, non-COVID-19 respiratory. But health is more than these two aspects. So I think this statement is misleading.
- No DCWs were recorded as dissenting for their registration data to be made available for linkage. I do not understand that sentence. Sorry.

Introduction

You wrote: Similarly, in England, most DCWs are female (84%), and work part-time (54%), about half work on zero hours contracts

(48%) (the highest proportion of workers in adult social care), are aged on average 44 years old, and staff turnover is high (35%). What is meant by "work on zero hours contracts"?

You wrote: During the pandemic, in England, roles in domiciliary care increased by around 7.4% compared to those in care homes, which observed a 0.2% decrease [3].

What is meant by "roles in domiciliary care"?

You wrote: However, the survey did not aim to capture objectively recorded health outcomes specifically in DCWs, for example positive rates of COVID-19 infection and the risk factors which contribute to these.

But why is it important to know this? in this specific group of health care workers? Is there a hypothesis, that they might be more often affected by COVID, as they work in close contact with the clients at home and not in a "safe" area such as a hospital or long-term care institution? Please explain here in detail, the research gap.

MATERIALS AND METHODS: Study design and data sources
You wrote: are held by Social Care Wales (DSCW) I think this is the wrong abbreviation, or?

You wrote: Prior to data being transferred to SAIL Databank from SCW, all DCWs had the opportunity to opt-out of their data going into the SAIL Databank for use in research.

Underneath you wrote: The study population was all registered DCWs resident in Wales on 1st March 2020 who did not subsequently opt-out to their data being linked for research,... I think you can delete one of it, as it says the same.

You wrote: The study population was all registered DCWs....underneath you wrote: individuals not registered as DCWs I think you can delete one of it, as it says the same.

You wrote: The primary outcome was confirmed COVID-19 infection, defined as the earliest of the following events: a positive polymerase chain reaction (PCR) test, hospital admission, death registration from COVID- 19, or a COVID-19 diagnosis Read code. When I read this, then I would interpret the sentence, that a positive polymerase chain reaction was one a confirmed COVID-19 infection and the primary outcome. Am I right? And what are the other variables namely hospital admission, death registration from COVID- 19, or a COVID-19 diagnosis Read code? Were these also the primary outcome as a confirmed COVID-19 infection? Or were these other variables additional important primary outcomes?

You wrote: The primary outcome was ... or a COVID-19 diagnosis Read code. What is a Read Code?

You wrote: Fit notes as a general marker of medically confirmed (return from) illness What do you mean by that?

	You wrote: Non-COVID-19 hospital admissions for respiratory conditions were examined for potential miscoding of COVID-19 infection in the early pandemic and included any lower respiratory infection, pneumonia, and influenza-like illness [17], exacerbations of chronic obstructive pulmonary disease (COPD) [18] and asthma [19]. If I understood you right, you checked the hospital admissions in the early pandemic. What do you mean by early pandemic? Furthermore, if I understood you right, you checked each lower respiratory infection, pneumonia, influenza-like illness, COPD and asthma if it really was a non-COVID-19 hospital admissions. Who did that? How could you be sure, that this was correct? You wrote: DCWs were followed up until the earliest of occurrence of death, first migration out of Wales, the outcome of interest, or the end of the follow-up. Ok I understand that if the person died or left Wales, that you cannot follow up anymore. However, what is if the person had Covid-19 in the 1, and the 2. Wave. You would have stopped the follow after the first wave. Did I understand you correctly? If yes, don't you think this is a limitation, as Covid-19 can occur several times? MATERIALS AND METHODS: Analysis You wrote: The prevalence estimates of outcomes were generalised to other nations. I think this does not belong to the analysis section. Especially, because I cannot find in the result section something about your outcomes, that were generalised to other nations. MATERIALS AND METHODS: Patient and public involvement You wrote: Two stakeholder groups provided input to the project, an ongoing Study Advisory Group and an Implementation Reference Group. Both groups included membership drawn from the domiciliary care sector. The Advisory Group met four times and provided scientific and policy oversight, monitored study progress and contributed to interpretation of emerging results. Membership included representation from a care workers charity, the care sector regulator and a service user. The Implementation Reference Group met twice and contributed to discussions about emerging findings, developing recommendations for policy and practice and advised on implementation routes and modes. Its membership included representation from care workers (including care manager), care users, trade unions, regulator, sector skills council and policy. Two policy briefings were co-produced with input from the IRG membership. From my point of view, this is too much information focusing only on that manuscript. I would only include: Two stakeholder groups provided input to the project, an ongoing Study Advisory Group and an Implementation Reference Group. Both groups included membership drawn from the domiciliary care sector. The members of both groups contributed to interpretation of the emerging results, with focus on that study.
--	---

Results: Study population

You wrote: Data from 15,931 DCWs resident and working in Wales on 1st March 2020 were linked to EHR data (85% of them had GP records in SAIL) with 15,725 (98.7%) DCWs included in the final analysis. I was wondering, why did you exclude 206 DCWs (15,931-15,725=206)?

You wrote in your aim in the introduction: Specifically we aimed to quantify rates of suspected and confirmed COVID-19 infection and key health outcomes as a consequence of COVID-19, explore outcome variation by demographics, workrelated factors, lifestyle and comorbidities and over time, and examine how generalisable the quantitative findings were to other UK nations. I would prefer, if you would order the results also in that way. Please also be aware, that you e.g. did not mention "Mental health" or "fit notes" as secondary outcomes in the aim.

Results: Confirmed COVID-19 infection

You wrote: Confirmed COVID-19 infection was 1.2% at 31st August 2020, rose to 14% by the end of February 2021, and reached 24% by the end of November 2021 (Table 2). I was wondering, if you also have the numbers, on how often 1,2,3,4 times one person was infected during the study period. This would be very interesting too.

After adjusting for age, sex and deprivation, COVID-19 infection was still more prevalent in those with a co-morbidity (HR=1.08, 95% CI: 1.01 to 1.15) and additionally adjusting for health board, rurality, and qualification type, was more prevalent in those employed by the Local Authority Social Services at registration (1.35, 1.23 to 1.47). I cannot find the numbers in one of the tables. So could you please explain, how and why you came to that numbers.

Table 1:

On COVID-19 shielding list: What is meant by that?
WIMD 2019 quintile: What is meant by that?

Results: Mental Health

You wrote: prescribing was 13% in March 2016 rising to 20% in March 2020 (Figure 2a), increasing on average per month by 17.61 (95% CI: 15.51 to 19.70, p-value <0.001). No evidence of a change was observed immediately following the start of the pandemic (March to April 2020) but evidence of an increase after April 2020 compared to the pre-pandemic with an average monthly increase of 29.34 (20.71 to 37.97, <0.001). Similar

patterns were observed for GP consultations. Mental health related hospital admissions were lower in March 2016 (around 15 admission per month) and significantly increased on average per month by 0.15 (0.06 to 0.25, 0.002) (Figure 2b). There was evidence to suggest a decrease in admissions immediately following the start of the pandemic but on increased on average by 0.99 admissions per month (0.44 to 1.53, <0.001), returning to pre-pandemic by the end of the study period.

I think these are very interesting results. However, you did not mention in the introduction, or in your aim, that you also compare the findings on mental health, with the pre-pandemic situation. Therefore, I also miss an explanation, why the comparison is important in the introduction. Please elaborate on that in the introduction.

Results: Suspected COVID-19 infection

You wrote: Flagged suspected COVID-19 infection.... What is meant by that?

Results: Generalisability

I would prefer, if the heading would be something like „Comparison to other UK nations”.

Discussion: Strengths and weaknesses

You wrote: A principal strength of the study was the comprehensive and contemporaneous coverage of the care worker population in Wales through linkage to registration data. I think this should be domiciliary care workers, or?

You wrote: The registration process was introduced ... and approaches to mitigation. Could you rewrite the sentence, so that it is clear for a non- English native speaker, if you think that this is a weakness of the databases.

You wrote: The Administrative records do not usually ...and wellbeing consequences of the pandemic for care workers. I agree, with your argumentation, when talking about medical diagnosis e.g. COVID-19 infection, mental health diagnosis, or even fit notes. However, your data did not focus on well-being, so this is (1) not comparable and (2) might be a limitation. I would rephrase this, as your study results adds to the already existing evidence to develop a more complete picture of the health and wellbeing consequences of the pandemic for care workers.

You wrote: It remains to be determined whether any facet of the employment role such as staff training, occupational risk assessment or testing procedures may have contributed these

differences or provide an opportunity to intervene. I think this is a conclusion and implication for further research.

Findings in context

You wrote: ONS data from the initial three-month period of the pandemic suggested mortality rates for individuals employed as care workers and home carers exceeded that found in the general population [30].

Did I understand you correctly?

Our results based on the ONS data from the initial three-month period of the pandemic suggested mortality rates for individuals employed as DCWs exceeded that found in the general population [30].

You wrote: However, data on occupational classification was missing for a large proportion of males and females dying with some COVID-19 involvement. even where available, data on occupation may have been incorrect due to role and job changes since census data were provided... And now I am confused. Because, if you did not have the occupational classification (or unsure that it might be correct), how could you know that these were DCWs?

You wrote: Our findings are more in line with the initial estimates provided by the Public Health England survey in June 2020 [31]. Very nice. Also please explain the similarities/differences between your results and the results of PH England.

You wrote: Our findings provide health outcome estimates over a much longer period, for a population rather than sample of care workers and for a broader range of outcomes. Very nice. However, in this paragraph it does not fit for me. I think you can mention that in the strength & limitation section.

You wrote: Despite role differences between the US sample and domiciliary care workers in Wales, the pre-pandemic levels of mental ill-health are strikingly similar. Can you please give the

	reader an explanation, why your results are similar with the US sample? You wrote: Evidence from our own work ...community of care workers may offer considerable long-term benefits to workforce and clients alike. For me this is a recommendation/implication for further research/policy. However, it does not fit for me in the "Findings in context" section. You wrote: Specifically we aimed to quantify rates of suspected and confirmed COVID-19 infection and key health outcomes as a consequence of COVID-19, explore outcome variation by demographics, workrelated factors, lifestyle and comorbidities and over time, and examine how generalisable the quantitative findings were to other UK nations. Whats in the "Findings in context" section.  • Key health outcomes: mental health [ ] YES • Key health outcomes: mortality [ ] YES Whats not in the "Findings in context" section:  • Suspected COVID-19 infection: • Confirmed COVID-19 infection: • Key health outcomes: fit notes • Explore outcome variation... • Examine generalizability to other UK nations From my point of view, especially a detailed discussion on your primary outcomes, should be included in the discussion section. Please work on that. Conclusion You wrote: Higher rates of confirmed COVID-19 infection amongst domiciliary care workers in Wales did not translate into mortality rates greater than for the broader Welsh population. High baseline rates of mental ill-health further increased over time, .. From my point of view these are results and not a conclusion.
--	--

VERSION 1 – AUTHOR RESPONSE

Reviewer: 1

R1.1 Really important paper and valuable contribution to understanding the impact of Covid-19 on this specific workforce.

RESPONSE: Thank you

R1.2 Line 6 - is dom care provided publicly funded by the state, or do people make co-payments, or is some of it privately funded - might be useful to include if known

RESPONSE: Domiciliary care can be funded by any of the means above. we have added a sentence to the first paragraph of the introduction:

“In the UK, domiciliary care can be funded publicly by local councils, privately funded, or a combination of both.”

R1.3 Line 27 - could it be explained how roles increased?

RESPONSE: Between 2019/20 and 2020/21 the number of available jobs in domiciliary care increased by 40,000 (a 7% increase). The vast majority of the increase in adult social care jobs was observed in CQC regulated domiciliary care services. Reference: The State of the Adult Social Care Sector and Workforce 2021 (skillsforcare.org.uk)

We have added this to paragraph 2 of the introduction:

*“During the pandemic, in England, **the number of available jobs** in domiciliary care increased by around 7.4% compared to those in care homes...”*

Reviewer: 2

(1) Please go more into depth in the introduction, why it is important to investigate

- your topic,
- this special sample,
- the different primary and secondary outcomes, as well as
- the comparison with other UK nations or
- the comparison with the pre pandemic situation.

I miss a clear argumentation, why this is important.

RESPONSE: We have updated the introduction to cover why this topic is important, in this specific workforce, the justification for the outcomes and the comparison to other UK nations and the pre-pandemic.

(2) Please stick to your aim including all variables, the order of the variables in the method, result as well as in the discussion section.

RESPONSE: We have included all outcomes in the aim and kept to the same order as in the results and discussion.

(3) Please discuss the following outcome variables (Especially the outcome variables) also in the discussion section.

- Suspected COVID-19 infection:
- Confirmed COVID-19 infection:
- Key health outcomes: fit notes
- Explore outcome variation...
- Examine generalizability to other UK nations

RESPONSE: We have included all outcomes in the discussion section.

In order to be able to give a precise feedback, you can find my specific comments and questions, stated below in green.

RESPONSE: Thank you for your comments. We have addressed the points in detail below.

R2.1 Abstract: Objectives

You wrote: To quantify population health risks for domiciliary care workers in Wales working during the COVID-19 pandemic. In your aim at the end of the introduction section you wrote the following: The OSCAR study aimed...to understand the impact of COVID-19 on DCWs in Wales. So do you quantify something or do you want to understand the impact of the pandemic?

RESPONSE: We aimed to quantify the risks for domiciliary care workers. We have changed the Introduction to read as follows :

“ The OSCAR study (Outcomes for Social Carers: an Analysis using Routine data) aimed to utilise the registration data collected by Social Care Wales, individually linked to secure anonymised electronic health record (EHR data sources) to quantify population health risks for DCWs in Wales working during the COVID-19 pandemic.”

R2.2 Abstract: Primary and secondary outcome measures

You wrote: Our primary outcome was confirmed COVID-19 infection; secondary outcomes included contacts for mental health, fit notes, respiratory infections not necessarily recorded as COVID-19, deaths involving COVID-19, all-cause mortality and suspected COVID-19.

In your method section you wrote the following: The primary outcome was confirmed COVID-19 infection, defined as the earliest of the following events: a positive polymerase chain reaction (PCR) test, hospital admission, death registration from COVID- 19, or a COVID-19 diagnosis Read code. So what is with , hospital admission, death registration from COVID- 19, or a COVID-19 diagnosis Read code as primary outcome?

RESPONSE: The primary outcome uses all the data sources i.e. polymerase chain reaction (PCR) data, hospital admissions, GP consultations, and deaths to ascertain the earliest contact indicating a confirmed COVID-19 infection.

R2.3 And I cannot find the numbers for deaths involving COVID-19 in your result section. Sorry. You only wrote once: The mortality rate amongst care workers was no greater than that observed amongst the general population of 15- to 64-year-olds in Wales (which was 0.034%). But here it is not stated that it is deaths involving COVID-19.

RESPONSE: we have added the COVID-19 related deaths to the abstract and results now reads:

“All-cause and COVID-19 related mortality are no greater than for the general population of 15- to 64-year-olds in Wales (0.1% and 0.034% respectively).”

We also realised that we have omitted a section relating to all-cause and covid-19 related mortality results. This has now been added to a section headed 'Deaths' and reads:

“Both all-cause and COVID-19-related mortality involved 0.1% and 0.034% of the study population respectively over the long-term study period (1st March 2020 and 30th November 2021).”

R2.4 STRENGTHS AND LIMITATIONS OF THIS STUDY

You wrote:

- Relatively little is known about the objective health of the UK's domiciliary care workforce (DCWs), either pre- or during the COVID-19 pandemic; our study aims to address this deficit. Interesting, as there is no information about the pre-pandemic situation except for Mental health illness, non-COVID-19 respiratory. But health is more than these two aspects. So I think this statement is misleading.

RESPONSE: We have removed this from the list of Strengths and limitations of this study

R2.5 You wrote: No DCWs were recorded as dissenting for their registration data to be made available for linkage.

I do not understand that sentence. Sorry.

RESPONSE: We have changed this the point under 'Strengths and Limitations of this Study' to read:

"No DCWs dissented for their registration data to be made available for linkage to electronic health records."

R2.6 Introduction

You wrote: Similarly, in England, most DCWs are female (84%), and work part-time (54%), about half work on zero hours contracts (48%) (the highest proportion of workers in adult social care), are aged on average 44 years old, and staff turnover is high (35%). What is meant by "work on zero hours contracts"?

RESPONSE: A 'Zero hours contract' is a non-legal term used to describe many different types of casual agreements between an employer and an individual and in which the employer does not guarantee the individual any hours of work.

R2.7 You wrote: During the pandemic, in England, roles in domiciliary care increased by around 7.4% compared to those in care homes, which observed a 0.2% decrease [3].

What is meant by "roles in domiciliary care"?

RESPONSE: this sentence has now been changed for clarity and now reads:

*"During the pandemic, in England, **the number of available jobs** in domiciliary care increased by around 7.4% compared to those in care homes, which observed a 0.2% decrease [3]."*

R2.8 You wrote: However, the survey did not aim to capture objectively recorded health outcomes specifically in DCWs, for example positive rates of COVID-19 infection and the risk factors which contribute to these.

But why is it important to know this? in this specific group of health care workers? Is there a hypothesis, that they might be more often affected by COVID, as they work in close contact with the

clients at home and not in a "safe" area such as a hospital or long-term care institution? Please explain here in detail, the research gap.

RESPONSE: We have added a sentence to explain why we hypothesise that domiciliary care workers may see an increased exposure to COVID-19 infection. The Introduction now reads:

"The close working conditions with clinically vulnerable people, with varied functions that DCWs perform in a non-institutional setting in a workforce with a large number of employers may increase exposure. A cross-sectional survey led by Ulster University..."

R2.9 MATERIALS AND METHODS: Study design and data sources

You wrote: are held by Social Care Wales (DSCW) I think this is the wrong abbreviation, or?

RESPONSE: We have clarified that the dataset is called the Domiciliary Social Care Worker (DSCW) dataset, and the sentence now reads:

"Registration data for all DCWs in Wales registered by 1st April 2020 are held by Social Care Wales (Domiciliary Social Care Worker (DSCW))"

R2.10 You wrote: Prior to data being transferred to SAIL Databank from SCW, all DCWs had the opportunity to opt-out of their data going into the SAIL Databank for use in research. Underneath you wrote: The study population was all registered DCWs resident in Wales on 1st March 2020 who did not subsequently opt-out to their data being linked for research,... I think you can delete one of it, as it says the same.

RESPONSE: We have deleted the sentence:

"Prior to data being transferred to SAIL Databank from SCW, all DCWs had the opportunity to opt-out of their data going into the SAIL Databank for use in research."

We also changed the following sentence:

"The study population was all registered DCWs resident in Wales on 1st March 2020 who did not subsequently opt-out to their data being transferred to the SAIL Databank and linked for research, either when DSCW data were added to the SAIL Databank or via their general practice (GP)."

R2.11 You wrote: The study population was all registered DCWs....underneath you wrote: individuals not registered as DCWs I think you can delete one of it, as it says the same.

RESPONSE: We have amended this sentence to be clearer and now reads:

"Additionally, individuals included in the dataset but registered as Domiciliary Care Managers, Adult Care Home Managers or Residential Child Care Workers were excluded."

R2.12 You wrote: The primary outcome was confirmed COVID-19 infection, defined as the earliest of the following events: a positive polymerase chain reaction (PCR) test, hospital admission, death registration from COVID- 19, or a COVID-19 diagnosis Read code. When I read this, then I would interpret the sentence, that a positive polymerase chain reaction was one a confirmed COVID-19 infection and the primary outcome. Am I right?

RESPONSE: Yes – the primary outcome of a confirmed COVID-19 infection could be detected by the presence of COVID-19 infection in any of these datasets e.g. presence in a GP consultation recorded with a diagnosis of covid-19 infection or a hospital admission recorded with a diagnosis of covid-19 infection, or a PCR test, or a death registration.

R2.13 And what are the other variables namely hospital admission, death registration from COVID-19, or a COVID-19 diagnosis Read code? Were these also the primary outcome as a confirmed COVID-19 infection? Or were these other variables additional important primary outcomes?

RESPONSE: All of these events were used to construct the primary outcome.

R2.14 You wrote: The primary outcome was ... or a COVID-19 diagnosis Read code. What is a Read Code?

RESPONSE: Read Codes are a coded thesaurus of clinical terms, used in the NHS since 1985. Read codes provide a standard vocabulary for clinicians to record patient findings and procedures, in health and social care IT systems across primary and secondary care. We have deleted the term from the sentence as it did not help the understanding of the outcome.

R2.15 You wrote: Fit notes as a general marker of medically confirmed (return from) illness What do you mean by that?

RESPONSE: A fit note is an official written statement from a registered healthcare professional giving their medical opinion on a person's fitness for work. On reflection, we have deleted the phrase "return from" in both the bullet point and in the results section 'Fit notes':

"Fit notes as a general marker of medically confirmed illness (Data source: WLGP)"

and

"The issuing of fit notes as a general marker of medically confirmed illness, increased over the time-period from 5% in August 2020,"

2.16 You wrote: Non-COVID-19 hospital admissions for respiratory conditions were examined for potential miscoding of COVID-19 infection in the early pandemic and included any lower respiratory infection, pneumonia, and influenza-like illness [17], exacerbations of chronic obstructive pulmonary disease (COPD) [18] and asthma [19].

If I understood you right, you checked the hospital admissions in the early pandemic. What do you mean by early pandemic?

RESPONSE: The early pandemic is defined by the period March to May 2020, where there was the potential of miscoding COVID-19 infection as other respiratory infections.

2.17 Furthermore, if I understood you right, you checked each lower respiratory infection, pneumonia, influenza-like illness, COPD and asthma if it really was a non-COVID-19 hospital admissions. Who did that? How could you be sure, that this was correct?

RESPONSE: We didn't check that each admission was a non-COVID-19 infection. We described the monthly hospital admissions recorded as either lower respiratory infection, pneumonia, and influenza-like-illness, exacerbations of chronic obstructive pulmonary disease (COPD) or asthma. The hypothesis was that if there was misclassification of infections occurring, we would expect the admission rate in these areas to increase in comparison to the rates seen pre-March 2020. The hypothesis was tested using an interrupted time series approach. Wee have changed the appropriate paragraph in the Analysis section to read:

"To explore changes over time and the effect of the COVID-19 pandemic period, an interrupted time series approach was used to test the hypothesis that hospital admissions for non-COVID-19 respiratory infections would increase in comparison to the rates seen pre-March 2020 if there was misclassification of infections occurring."

2.18 You wrote: DCWs were followed up until the earliest of occurrence of death, first migration out of Wales, the outcome of interest, or the end of the follow-up. Ok I understand that if the person died or left Wales, that you cannot follow up anymore. However, what is if the person had Covid-19 in the 1, and the 2. Wave. You would have stopped the follow after the first wave. Did I understand you correctly? If yes, don't you think this is a limitation, as Covid-19 can occur several times?

RESPONSE: Yes this is correct. All our analyses examined time to **first event** as this was the main outcome of interest. We did not consider recurrent events.

2.19 MATERIALS AND METHODS: Analysis

You wrote: The prevalence estimates of outcomes were generalised to other nations. I think this does not belong to the analysis section. Especially, because I cannot find in the result section something about your outcomes, that were generalised to other nations.

RESPONSE: Our aim was to examine how generalisable findings in relation to confirmed COVID-19 were to other UK nations. The results of this exercise can be found in the last paragraph of the results section under "Comparison to other UK nations".

R2.20 MATERIALS AND METHODS: Patient and public involvement

You wrote: Two stakeholder groups provided input to the project, an ongoing Study Advisory Group and an Implementation Reference Group. Both groups included membership drawn from the domiciliary care sector. The Advisory Group met four times and provided scientific and policy oversight, monitored study progress and contributed to interpretation of emerging results.

Membership included representation from a care workers charity, the care sector regulator and a service user. The Implementation Reference Group met twice and contributed to discussions about emerging findings, developing recommendations for policy and practice and advised on implementation routes and modes. Its membership included representation from care workers (including care manager), care users, trade unions, regulator, sector skills council and policy. Two policy briefings were co-produced with input from the IRG membership. From my point of view, this is too much information focusing only on that manuscript. I would only include:

Two stakeholder groups provided input to the project, an ongoing Study Advisory Group and an Implementation Reference Group. Both groups included membership drawn from the domiciliary care sector. The members of both groups contributed to interpretation of the emerging results, with focus on that study.

RESPONSE: We have shortened this paragraph as recommended and it now reads as follows:

“Two stakeholder groups provided input to the project, an ongoing Study Advisory Group and an Implementation Reference Group. Both groups included membership drawn from the domiciliary care sector and both contributed to interpretation of emerging results.”

R2.21 Results: Study population

You wrote: Data from 15,931 DCWs resident and working in Wales on 1st March 2020 were linked to EHR data (85% of them had GP records in SAIL) with 15,725 (98.7%) DCWs included in the final analysis. I was wondering, why did you exclude 206 DCWs (15,931-15,725=206)?

RESPONSE: 206 DCWs were excluded due to matching errors. We have added a line to state this:

“Records from 206 DCWs were excluded due to matching errors, leaving 15,725 (98.7%) DCWs included in the final analysis”

R2.22 You wrote in your aim in the introduction: Specifically we aimed to quantify rates of suspected and confirmed COVID-19 infection and key health outcomes as a consequence of COVID-19, explore outcome variation by demographics, work related factors, lifestyle and comorbidities and over time, and examine how generalisable the quantitative findings were to other UK nations. I would prefer, if you would order the results also in that way. Please also be aware, that you e.g. did not mention “Mental health” or “fit notes” as secondary outcomes in the aim.

RESPONSE: We have clarified the key outcomes including mental health, fit notes, respiratory infections not necessarily recorded as COVID-19, deaths involving COVID-19, and all-cause mortality.

The results are now presented as listed under the section **Primary and secondary outcomes** with the Primary outcome (Confirmed COVID-19) first, and then the secondary outcomes:

- Suspected COVID-19 infection;
- Contacts for mental health and diagnoses, psychotropic medication and admissions (including self-harm);
- Fit notes as a general marker of medically confirmed illness;
- Deaths involving COVID-19 and all-cause mortality.

The exception to this are the results for non-COVID-19 hospital admissions for respiratory condition, that need to be discussed alongside the results for confirmed COVID-19, so that the potential of miscoding of COVID-19 infection can be understood.

R2.23 Results: Confirmed COVID-19 infection

You wrote: Confirmed COVID-19 infection was 1.2% at 31st August 2020, rose to 14% by the end of February 2021, and reached 24% by the end of November 2021 (Table 2). I was wondering, if you also have the numbers, on how often 1,2,3,4 times one person was infected during the study period. This would be very interesting too.

RESPONSE: Unfortunately we do not have these results. As COVID-19 was measured by a composite measure from different data sources, ensuring that each record was a new infection and not related would have been challenging.

R2.24 After adjusting for age, sex and deprivation, COVID-19 infection was still more prevalent in those with a co-morbidity (HR=1.08, 95% CI: 1.01 to 1.15) and additionally adjusting for health board, rurality, and qualification type, was more prevalent in those employed by the Local Authority Social Services at registration (1.35, 1.23 to 1.47). I cannot find the numbers in one of the tables. So could you please explain, how and why you came to that numbers.

RESPONSE: The adjusted odds ratio and 95% confidence intervals (resulting from a multivariable time to event regression model with comorbidity as the independent variable and COVID-19 infection as the dependent variable) is only presented in the text and not in the table. Similarly with the independent variable, employment sector.

R2.25 Table 1: On COVID-19 shielding list: What is meant by that?

RESPONSE: The COVID-19 shielded patient list included people that were at a higher risk of getting seriously ill from COVID-19 (e.g. people with cancer, sickle cell disease) and were advised to take extra steps or follow additional advice to protect yourself from COVID-19; to 'shield'. We have added a footnote to Table 1 to explain:

“people at a higher risk of getting seriously ill from COVID-19”

R2.26 WIMD 2019 quintile: What is meant by that?

RESPONSE: WIMD stands for the Welsh Index of Multiple Deprivation and is designed to identify the small areas of Wales that are the most deprived by attributing a score to each small area. Quintiles are derived by proportioning the deprivation score into five equal groups. This explanation has been added to Table 1 itself and in a footnote:

“The Welsh Index of Multiple Deprivation (WIMD) is the official measure of relative deprivation for small areas in Wales. Quintiles are derived by proportioning the deprivation score into five equal groups.”

R2.27 Results: Mental Health

You wrote: prescribing was 13% in March 2016 rising to 20% in March 2020 (Figure 2a), increasing on average per month by 17.61 (95% CI: 15.51 to 19.70, p-value <0.001). No evidence of a change was observed immediately following the start of the pandemic (March to April 2020) but evidence of an increase after April 2020 compared to the pre-pandemic with an average monthly increase of 29.34 (20.71 to 37.97, <0.001). Similar patterns were observed for GP consultations. Mental health related hospital admissions were lower in March 2016 (around 15 admission per month) and significantly increased on average per month by 0.15 (0.06 to 0.25, 0.002) (Figure 2b). There was evidence to suggest a decrease in admissions immediately following the start of the pandemic but on increased on average by 0.99 admissions per month (0.44 to 1.53, <0.001), returning to pre-pandemic by the end of the study period.

I think these are very interesting results. However, you did not mention in the introduction, or in your aim, that you also compare the findings on mental health, with the pre-pandemic situation. Therefore, I also miss an explanation, why the comparison is important in the introduction. Please elaborate on that in the introduction.

RESPONSE: We have added the justification of this analysis to the introduction and included it as an aim:

“We compared mental health outcomes within the pandemic period to the pre-pandemic situation to assess the effect of the pandemic”

R2.28 Results: Suspected COVID-19 infection

You wrote: Flagged suspected COVID-19 infection.... What is meant by that?

RESPONSE: Flagged suspected covid-19 infections indicated domiciliary care workers that were noted as having a relevant code recorded in electronic health records. We have omitted the word 'Flagged' from the sentence and it now reads:

“Examining the prevalence of suspected COVID-19 infection added little to the clinical picture”.

R2.29 Results: Generalisability

I would prefer, if the heading would be something like “Comparison to other UK nations”.

RESPONSE: We have replaced this title as requested.

R2.30 Discussion: Strengths and weaknesses

You wrote: A principal strength of the study was the comprehensive and contemporaneous coverage of the care worker population in Wales through linkage to registration data. I think this should be domiciliary care workers, or?

RESPONSE: We have changed this to:

*“A principal strength of the study was the comprehensive and contemporaneous coverage of the **domiciliary care worker population** in Wales through linkage to registration data.”*

R2.31 You wrote: The registration process was introduced ... and approaches to mitigation. Could you rewrite the sentence, so that it is clear for a non- English native speaker, if you think that this is a weakness of the databases.

RESPONSE: We have re-written this paragraph in the Strengths and weaknesses section.

R2.32 You wrote: The Administrative records do not usually ...and wellbeing consequences of the pandemic for care workers. I agree, with your argumentation, when talking about medical diagnosis e.g. COVID19 infection, mental health diagnosis, or even fit notes. However, your data did not focus on wellbeing, so this is (1) not comparable and (2) might be a limitation. I would rephrase this, as your study results adds to the already existing evidence to develop a more complete picture of the health and wellbeing consequences of the pandemic for care workers.

RESPONSE: We have taken out the reference to 'wellbeing' and the sentence now reads:

“Findings from our routine data study builds a more complete picture of the health consequences of the pandemic for care workers.”

R2.34 You wrote: It remains to be determined whether any facet of the employment role such as staff training, occupational risk assessment or testing procedures may have contributed these differences or provide an opportunity to intervene. I think this is a conclusion and implication for further research.

RESPONSE: We have moved this sentence into the Conclusion section.

R2.35 Findings in context

You wrote: ONS data from the initial three-month period of the pandemic suggested mortality rates for individuals employed as care workers and home carers exceeded that found in the general population [30]. Did I understand you correctly? Our results based on the ONS data from the initial three-month period of the pandemic suggested mortality rates for individuals employed as DCWs exceeded that found in the general population [30].

RESPONSE: This sentence suggests that previous research conducted by the ONS using ONS data, suggested that care workers and home carers had a higher mortality rate during the early pandemic, when compared to the general population. We have amended the sentence for clarity :

“ONS data from the initial three-month period of the pandemic suggested that mortality rates for individuals employed as care workers and home carers were higher than those found in the general population [30].”

R2.36 You wrote: However, data on occupational classification was missing for a large proportion of males and females dying with some COVID-19 involvement. even where available, data on occupation may have been incorrect due to role and job changes since census data were provided... And now I am confused. Because, if you did not have the occupational classification (or unsure that it might be correct), how could you know that these were DCWs?

RESPONSE: The missing/potentially inaccurate data was in relation to the data reported by the ONS and not our study.

R2.37 You wrote: Our findings are more in line with the initial estimates provided by the Public Health England survey in June 2020 [31]. Very nice. Also please explain the similarities/differences between your results and the results of PH England.

RESPONSE: We have added more information:

“Our findings are more in line with the initial estimates provided by a Public Health England survey of DCWs in July 2020, where the rate of confirmed COVID-19 on PCR testing was low (0.1% (95% CI=0.02 to 0.04%) [31].”

R2.38 You wrote: Our findings provide health outcome estimates over a much longer period, for a population rather than sample of care workers and for a broader range of outcomes. Very nice. However, in this paragraph it does not fit for me. I think you can mention that in the strength & limitation section.

RESPONSE: We have taken this sentence out of the Findings in context section of the Discussion and added this as a strength to the Strengths and limitations of this study:

“Our study provides health outcome estimates for DCWs over a longer period of the pandemic, for a population of care workers and for a broad range of outcomes.”

R2.39 You wrote: Despite role differences between the US sample and domiciliary care workers in Wales, the pre-pandemic levels of mental ill-health are strikingly similar. Can you please give the reader an explanation, why your results are similar with the US sample?

RESPONSE: The reason for such similarities are not know and not a focus of the current study. We have therefore added the following statement:

“The reasons for such similarities are unclear and could simply reflect broader population similarities in prevalence of mental ill-health rather than occupationally driven factors.”

R2.40 You wrote: Evidence from our own work ...community of care workers may offer considerable longterm benefits to workforce and clients alike. For me this is a recommendation/implication for further research/policy. However, it does not fit for me in the “Findings in context” section.

RESPONSE: As the BMJ open does not include a recommendations section, we have included this under our conclusions section.

R2.41 You wrote: Specifically we aimed to quantify rates of suspected and confirmed COVID-19 infection and key health outcomes as a consequence of COVID-19, explore outcome variation by demographics, workrelated factors, lifestyle and comorbidities and over time, and examine how generalisable the quantitative findings were to other UK nations.

Whats in the “Findings in context” section.

- Key health outcomes: mental health → YES
- Key health outcomes: mortality → YES

Whats not in the “Findings in context” section:

- Suspected COVID-19 infection:
- Confirmed COVID-19 infection:
- Key health outcomes: fit notes
- Explore outcome variation...
- Examine generalizability to other UK nations

From my point of view, especially a detailed discussion on your primary outcomes, should be included in the discussion section. Please work on that.

RESPONSE: We have added a detailed discussion on COVID-19 infection to the Findings in context section of the Discussion.

R2.42 Conclusion

You wrote: Higher rates of confirmed COVID-19 infection amongst domiciliary care workers in Wales did not translate into mortality rates greater than for the broader Welsh population. High baseline rates of mental ill-health further increased over time, .. From my point of view these are results and not a conclusion.

RESPONSE: We believe that an overview of the findings from the study should be included in the conclusion. We have however added to the section, as recommended under point R2.40, and this now reads as follows:

“This study presents evidence of the direct and indirect impact of COVID-19 pandemic upon DCWs in Wales. Higher rates of confirmed COVID-19 infection amongst DCWs did not translate into mortality rates greater than for the broader Welsh population. High baseline rates of mental ill-health further increased over time, a burden that fell unevenly across the workforce. Evidence from our own work and that of others, supports the value of co-produced solutions which draw on the direct experiences of care workers to support occupational related well-being [41]. Systemic drivers (e.g. public funding for social care, staffing levels, levels of pay) and situational aspects of the role such as peripatetic working will not change quickly or even at all. With few evidence-based supportive approaches tailored to the circumstances of care workers [42] [43] optimising or innovating approaches to support the UK community of care workers may offer considerable long-term benefits to workforce and clients alike.”

VERSION 2 – REVIEW

REVIEWER	Hoedl, Manuela Medical University of Graz, Institute of Nursing Science
REVIEW RETURNED	27-Apr-2023

GENERAL COMMENTS	Thank you for giving me the possibility to review your manuscript revision regarding the “The impact of the COVID-19 pandemic on Domiciliary Care Workers in Wales (UK): a data linkage cohort study using the SAIL Databank”. I think the manuscript really improved through the revision process. Nevertheless, there are still main aspects, from my point of view, that are missing in this manuscript. My biggest concern is, that you did not explain in the introduction section, why this study is important? Why did you do that. Just because you had the data is not a reason. So please go more into depth in the introduction, why it is important to investigate  • your topic, • this special sample, • the different primary and secondary outcomes, as well as
---

- the comparison with other UK nations or
- the comparison with the pre pandemic situation.

There is no clear rationale/research gap why you have conducted the study.

I also had a look again at the outcomes and your aims. Underneath you can find two tables. I hope the tables help you to organize the paper a little bit more.

- (1) Please stick to your aim including all variables, the order of the variables in the method, result as well as in the discussion section.
- (2) Please discuss the following outcome variables (Especially the outcome variables) also in the discussion section.
 - Suspected COVID-19 infection:
 - Confirmed COVID-19 infection:
 - Key health outcomes: fit notes
 - Explore outcome variation...
 - Examine generalizability to other UK nations

OUTCOMES & were I found it:

	CO VID- 19 infe ctio n	me ntal he alth	fit no tes	respi ratory infe ctions	deat hs invol ving CO VID- 19	all- cau se morta lity	susp ected COVI D-19	Se lf harm	Comp arison UK
Abstract	X	X	X	X	X	X	X		
Introduc tion:aim	X	X	X	X	X	X	X		
Method	X	X	X	X	X	X	X		
Results	X	X	X		X	X	X		X
Table 2								X	
Principal findings	X	X	X			X			X
Results in context					X	X			

Please think about that and have a look if e.g. self-harm is really needed in table 2. Please also go again through the whole manuscript, to ensure, that the order of the outcomes is the same.

Aims & were I found it:

	quantify populatio n health risks	compare mental health outcome s between pandemi	examined admission s over time for respiratory infection**	Explore outcome variation** *
--	--	---	---	--

			c & pre-pandemic situation*			
Abstract	X					
Introduction:aim	X	X	X	X		

* to assess the effect of the pandemic.
 ** to identify the potential miscoding of COVID-19 infection.
 *** to develop timely public health policy messages

stated below in green.

Abstract: Results

You wrote: Suspected and confirmed COVID-19 rates...Please stick to the order of your aims/outcome variables.

STRENGTHS AND LIMITATIONS OF THIS STUDY

You wrote:

- No DCWs were recorded as dissenting for their registration data to be made available for linkage. I do not understand that sentence. Sorry.

Introduction

You wrote: However, the survey did not aim to capture objectively recorded health outcomes specifically in DCWs, for example positive rates of COVID-19 infection and the risk factors which contribute to these.

But why is it important to know this? in this specific group of health care workers? Is there a hypothesis, that they might be more often affected by COVID, as they work in close contact with the clients at home and not in a "safe" area such as a hospital or long-term care institution? Please explain here in detail, the research gap.

These were the aims of your study:

- to quantify rates of suspected and confirmed COVID-19 infection
- to establish adverse health outcomes affecting DCWs, I have no idea, what are these?
- to identify key health outcomes: mental health, fit notes, COVID-19 mortality
- to compared mental health outcomes to assess the effect of the pandemic.
- to identify the potential miscoding of COVID-19 infection.
- to explore outcome variation

I think here the comparison with other UK nations is missing? Or?

MATERIALS AND METHODS: Study design and data sources

You wrote: DCWs were followed up until the earliest of occurrence of death, first migration out of Wales, the outcome of interest, or the end of the follow-up. Ok I understand that if the person died or left Wales, that you cannot follow up anymore. However, what is if the person had Covid-19 in the 1, and the 2. Wave. You would have stopped the follow after

the first wave. Did I understand you correctly? If yes, don't you think this is a limitation, as Covid-19 can occur several times?

Results: Confirmed COVID-19 infection

You wrote: Confirmed COVID-19 infection was 1.2% at 31st August 2020, rose to 14% by the end of February 2021, and reached 24% by the end of November 2021 (Table 2). I was wondering, if you also have the numbers, on how often 1,2,3,4 times one person was infected during the study period. This would be very interesting too.

Discussion: Strengths and weaknesses

You wrote: The registration process was introduced ... and approaches to mitigation. Could you please rewrite the sentence? It is not completely clear, if you think this is a strength or weakness of the databases.

Findings in context

You wrote: ONS data from the initial three-month period of the pandemic suggested mortality rates for individuals employed as care workers and home carers exceeded that found in the general population [30].

Did I understand you correctly?

In contrast to our findings ONS data from the initial three-month period of the pandemic suggested mortality rates for individuals employed as DCWs were higher than those found in the general population [30]. If yes, I would include "in contrast to our findings"

You wrote: However, data on occupational classification was missing for a large proportion of dying with some COVID-19 involvement. even where available, data on occupation may have been incorrect due to role and job changes since census data were provided... Could you please make clear, were these data were missing? Either from your study or the ONS data. And you can also use it in 1 sentence why the results of your and the ONS data were different.

You wrote: Our findings are more in line with the initial estimates provided by the Public Health England survey in June 2020 [31]. Very nice. Also please explain the similarities between your results and the results of PH England.

Whats in the "Findings in context" section.

- Key health outcomes: mental health → YES
- Key health outcomes: mortality → YES
- Confirmed COVID-19 infection: → YES

Whats still not yet in the "Findings in context" section:

- Suspected COVID-19 infection:
- Key health outcomes: fit notes
- Explore outcome variation...
- Examine generalizability to other UK nations

From my point of view, the findings in context section could still be improved. However, I understand, if you run out of space.

You wrote: A semi-quantitative job exposure matrix (COVID-19-JEM) was developed to estimate the likelihood of workers becoming infected with SARS-CoV-2 in an occupational setting [37]. 'Home workers' and 'home carers' were assessed as scoring 14 on the

	matrix, indicating a relatively high risk of exposure due to characteristics inherent in their job. However, the threshold used (13+) would include half of the UK workforce and the matrix's factors for transmission risk and mitigation could clearly mask a heterogeneity of risk as suggested in our own qualitative work [38]. This has nothing to do with the results presented here. It is a further development. I would prefer to delete it.
--	--

VERSION 2 – AUTHOR RESPONSE

Thank you for giving me the possibility to review your manuscript revision regarding the “The impact of the COVID-19 pandemic on Domiciliary Care Workers in Wales (UK): a data linkage cohort study using the SAIL Databank”. I think the manuscript really improved through the revision process.

Nevertheless, there are still main aspects, from my point of view, that are missing in this manuscript.

My biggest concern is, that you did not explain in the introduction why this study is important? Why did you do that. Just because you had the data is not a reason. So please go more into depth in the introduction, why it is important to investigate

1. your topic,
2. this special sample,
3. the different primary and secondary outcomes, as well as
4. the comparison with other UK nations or
5. the comparison with the pre pandemic situation.

There is no clear rationale/research gap why you have conducted the study.

RESPONSE:

1&2. The introduction covers why the health of the DCWs are important to examine during the pandemic:

“These close working conditions with clinically vulnerable people, and the varied functions that DCWs perform in a non-institutional setting in a workforce with a large number of employers, may increase exposure to COVID-19. “

and where other studies had already found evidence of an adverse impact:

“...both wellbeing and quality of working life deteriorated over all occupations including Social Care.”

We also highlight what our study aimed to add to other previous research in this particular population:

“However, the survey did not aim to capture objectively recorded health outcomes specifically in DCWs, for example positive rates of COVID-19 infection and the risk factors which contribute to these.”

We have added to the second paragraph of Introduction the following text:

“In 2020, the Office for National Statistics (ONS) reported elevated mortality rates due to COVID-19 for ‘care workers and home carers’ compared to the general population in England and Wales.[ref] In June 2020, Public Health England reported COVID-19 infection rates for DCWs in line with general population rates.[ref] Methodological differences between the two studies may explain contrasting estimates and neither study offered population coverage for a well-defined cohort of DCWs. Therefore, despite a policy interest in understanding level of risk for a population of workers potentially at increased likelihood of exposure to COVID-19 (and consequently the risk they could pose to clients), there was uncertainty about rates of COVID-19 infection or any other health outcome.”

To clarify the uncertainty in evidence base at the start of the study and the rationale for our approach.

3. The primary and secondary outcomes are listed in full and in order:

“Specifically, we aimed to quantify rates of confirmed and suspected COVID-19 infection and, to fully establish the range of adverse health outcomes potentially affecting DCWs such as mental health including self-harm, issuing of fit notes, non-COVID-19 hospital admissions for respiratory conditions, and COVID-19 related and all-cause mortality, as a consequence of the pandemic.”

4. We have expanded the text to include the comparison with other UK nations :

“To develop timely public health policy messages that could be extended to DCWs in other nations of the UK, using up-to-date DCW workforce data from Scotland, Northern Ireland and England we compared these populations to the Welsh DCW workforce and assess generalisability of prevalence findings to each nation.”

5. We also have added detail on the comparison with the pre-pandemic:

“We compared mental health outcomes (GP contacts and prescriptions, and hospital admissions) within the pandemic period to the pre-pandemic situation to assess the effect of the pandemic. Similarly we examined admissions over time for respiratory infection in the early pandemic to identify the potential miscoding of COVID-19 infection.”

I also had a look again at the outcomes and your aims. Underneath you can find two tables. I hope the tables help you to organize the paper a little bit more.

(1) Please stick to your aim including all variables, the order of the variables in the method, result as well as in the discussion section.

(2) Please discuss the following outcome variables (Especially the outcome variables) also in the discussion section.

- Suspected COVID-19 infection
- Confirmed COVID-19 infection:
- Key health outcomes: fit notes
- Explore outcome variation...

- Examine generalizability to other UK nations

OUTCOMES & where I found it:

	COVI D-19 infection	Men tal health	Fit notes	Respira tory infections	Death s involving COVI D -19	All- cause mortality	Suspe cted COVID -19	Sel f harm	Compar ison UK
Abstract	X	X	X	X	X	X	X	2	5
Introductio n:aim	X	X	X	X	X	X	X	2	5
Method	X	X	X	X	X	X	X	1	5
Results	X	X	X	3	X	X	X	2	X
Table 2	4	4	4	4	4	4	4	X	NA
Principal findings	X	X	X	6	6	X	6	2	X
Findings in context	7	7	7	7	X	X	7	7	7

Please think about that and have a look if e.g. self harm is really needed in table 2.

Please also go again through the whole manuscript, to ensure, that the order of the outcomes is the same.

Aims & where I found it:

	Quantify population health risks	Compare mental health outcomes between pandemic & pre-pandemic situation*	Examined admissions over time for respiratory infection***	Explore outcome variation***
Abstract	X	7	7	7
Introduction:aim	X	X	X	X

*to assess the effect of the pandemic

** to identify the potential miscoding of COVID-19 infection

***to develop timely public health policy messages

RESPONSE:

1. In the Methods section, self harm is listed as an outcome within the mental health secondary outcome. We have also referenced how it was coded.
2. We now discuss self harm in each of the areas highlighted in the table above.
3. Respiratory infections are included in the results . As explained in our last response, the results for non-COVID-19 hospital admissions for respiratory condition, need to be discussed alongside the results for confirmed COVID-19, so that the potential of miscoding of COVID-19 infection can be understood and as such are under the 'Confirmed COVID-19 infection' heading.
4. All outcomes are shown in table 2, apart from COVID-19 related deaths as numbers were too small to present. Comparison with other nations results are not included in table 2 as they are not relevant.
5. Detail on the Comparison with UK nations, have now been added to these sections.
6. Principal findings: we have now ensured that all outcomes are covered in the principal findings.
7. In the Findings in Context section, COVID-19 infections, mental health and mortality are already discussed. As our Discussion section is now over 2000 words, so without unduly lengthening the paper further we have focused on main findings to set in context. Outcomes such as suspected COVID-19 and fit notes are less direct measures / indicators of COVID hence why we focus on confirmed COVID-19 infections. With little other comparative data for COVID-19 infection and mental ill health (i.e. from a similarly well-defined comprehensively covered DCW population) there is less to say regarding outcome variation (e.g. by sub-groups) and extending findings to other UK nations is being attempted specifically because there is no good data available.
8. The objective of “To quantify population health risks for domiciliary care workers in Wales working during the COVID-19 pandemic.”, is an overarching one that would encompass both the comparison of outcomes pre-pandemic vs pandemic, and exploring variation in outcomes. Regarding the “Examined admissions over time for respiratory infection” even though is an outcome, is more of coding check and not deemed appropriate to be stated as an aim. The aim now incorporates developing timely public health policy messages:

“To quantify population health risks for domiciliary care workers in Wales working during the COVID-19 pandemic and develop timely public health policy messages.”

In order to be able to give a precise feedback, you can find my specific comments and questions, stated below in green.

Abstract: Results

You wrote: Suspected and confirmed COVID-19 rates...Please stick to the order of your aims/outcome variables.

RESPONSE: We have moved the suspected COVID-19 results so that they are the first secondary outcome discussed.

STRENGTHS AND LIMITATIONS OF THIS STUDY

You wrote: • No DCWs were recorded as dissenting for their registration data to be made available for linkage. I do not understand that sentence. Sorry.

RESPONSE: This was changed since the last review and now reads:

“No DCWs dissented for their registration data to be made available for linkage allowing a comprehensive population level analysis.”

Introduction

You wrote: However, the survey did not aim to capture objectively recorded health outcomes specifically in DCWs, for example positive rates of COVID-19 infection and the risk factors which contribute to these. But why is it important to know this? in this specific group of health care workers? Is there a hypothesis, that they might be more often affected by COVID, as they work in close contact with the clients at home and not in a "safe" area such as a hospital or long-term care institution? Please explain here in detail, the research gap.

RESPONSE: We have now expanded the introduction as detailed in our earlier response (page 2).

These were the aims of your study:

- to quantify rates of suspected and confirmed COVID-19 infection
- to establish adverse health outcomes affecting DCWs, I have no idea, what are these?

RESPONSE: We think that this sentence is a bit confusing and we have clarified as follows:

“Specifically, we aimed to quantify rates of suspected and confirmed COVID-19 infection and, to fully establish the range of adverse health outcomes affecting DCWs such as mental health, fit notes, COVID-19 related and all-cause mortality, as a consequence of the pandemic.”

- to identify key health outcomes: mental health, fit notes, COVID-19 mortality
- to compared mental health outcomes to assess the effect of the pandemic.
- to identify the potential miscoding of COVID-19 infection.
- to explore outcome variation

I think here the comparison with other UK nations is missing? Or?

RESPONSE: We have clarified this and now reads:

“ Outcome variation was explored by demographics, work-related factors, lifestyle and comorbidities to develop timely public health policy messages that could be extended to DCWs in other nations of the UK. Additionally, using up-to-date DCW workforce data from Scotland, Northern Ireland and England we will compare these populations to the Welsh DCW workforce and assess generalisability of prevalence findings to each nation.”

MATERIALS AND METHODS: Study design and data sources

You wrote: DCWs were followed up until the earliest of occurrence of death, first migration out of Wales, the outcome of interest, or the end of the follow-up. Ok I understand that if the person died or left Wales, that you cannot follow up anymore. However, what is if the person had Covid-19 in the 1, and the 2. Wave. You would have stopped the follow after the first wave. Did I understand you correctly? If yes, don't you think this is a limitation, as Covid-19 can occur several times?

RESPONSE: We understand that an individual DCW could have multiple COVID-19 infections but the aim of this study (and as pre-specified in the analysis plan) was to examine the time to the first COVID-19 infection. Considering recurrent events of COVID-19 infection was not a research question we prioritised in this study, so as such don't see its omission as a limitation for what we saw as our aims and objectives.

Results: Confirmed COVID-19 infection

You wrote: Confirmed COVID-19 infection was 1.2% at 31st August 2020, rose to 14% by the end of February 2021, and reached 24% by the end of November 2021 (Table 2). I was wondering, if you also have the numbers, on how often 1,2,3,4 times one person was infected during the study period. This would be very interesting too.

RESPONSE: As explained in our response to the first round of reviewer's comments, we unfortunately do not have these results. As confirmed COVID-19 infection was a composite measure using different data sources, ensuring that each record was a new episode of infection and not related would have been challenging.

Discussion: Strengths and weaknesses

You wrote: The registration process was introduced ... and approaches to mitigation. Could you please rewrite the sentence? It is not completely clear, if you think this is a strength or weakness of the databases.

RESPONSE: We have clarified:

“Integrating qualitative interviews (which have been separately reported) with the quantitative routine data analyses has provided insight into the working experience and context of DCWs and how that may translate to variability in infection risk and mitigation. It has illuminated the heterogeneity of roles being undertaken by those registered under the umbrella role of DCW and therefore we would consider a strength.”

Findings in context

You wrote: ONS data from the initial three-month period of the pandemic suggested mortality rates for individuals employed as care workers and home carers exceeded that found in the general population [30]. Did I understand you correctly?

RESPONSE: Yes this is correct.

In contrast to our findings ONS data from the initial three-month period of the pandemic suggested mortality rates for individuals employed as DCWs were higher than those found in the general population [30]. If yes, I would include “in contrast to our findings”

RESPONSE: We have included this phrase.

You wrote: However, data on occupational classification was missing for a large proportion of dying with some COVID-19 involvement. even where available, data on occupation may have been incorrect due to role and job changes since census data were provided... Could you please make clear, were these data were missing? Either from your study or the ONS data. And you can also use it in 1 sentence why the results of your and the ONS data were different.

RESPONSE: This sentence refers only to data reported in the ONS study. Missingness of occupational classification in ONS data is as we describe because occupation is derived from census data. We have clarified that census data are updated every 10 years (hence why data may be out of date). Our study has complete data on job role (as it defines inclusion in this study cohort).

“However, data on occupational classification was missing for a large proportion dying with some COVID-19 involvement and, even where available, data on occupation may have been incorrect due to role and job changes since census data were provided (i.e. every ten years).”

You wrote: Our findings are more in line with the initial estimates provided by the Public Health England survey in June 2020 [31]. Very nice. Also please explain the similarities between your results and the results of PH England.

RESPONSE: We have added:

“The PHE study differed from our study in several ways. In the PHE study, DCWs were identified through purposive sampling of care providing organisations by region (i.e. to achieve similar numbers of respondents by region), initially approached based on convenience. Within participating organisations, a convenience sample of staff were approached to take part in the survey which involved staff returning a nasal swab and short questionnaire during a two-week study window (2nd to 16th June, 2020). DCW staff included in the PHE study were 84.3% female, with a median age of 41 years and 75.8% of those providing details about ethnicity were white. Two of the 2,015 DCWs returning a swab in the two-week study period were positive for SARS-CoV-2 on PCR testing, while 41 DCWs reported symptoms of COVID-19 in the 14 days prior to the swab being taken.”

Whats in the “Findings in context” section

- Key health outcomes: mental health →YES
- Key health outcomes: mortality →YES
- Confirmed COVID-19 infection: →YES

Whats still not yet in the “Findings in context” section:

- Suspected COVID-19 infection:
- Key health outcomes: fit notes
- Explore outcome variation...
- Examine generalizability to other UK nations

From my point of view, the findings in context section could still be improved. However, I understand, if you run out of space.

RESPONSE: As we noted earlier, our Discussion section is now very lengthy so without unduly lengthening the paper further we have focused on main findings to set in context. Outcomes such as suspected COVID-19 and fit notes are less direct measures / indicators of COVID hence why we focus on confirmed COVID-19 infections. With little other comparative data for COVID-19 infection and mental ill health (i.e. from a similarly well-defined comprehensively covered DCW population) there is less to say regarding outcome variation (e.g. by sub-groups) and extending findings to other UK nations is being attempted specifically because there is no good data available.

You wrote: A semi-quantitative job exposure matrix (COVID-19-JEM) was developed to estimate the likelihood of workers becoming infected with SARS-CoV-2 in an occupational setting [37]. ‘Home workers’ and ‘home carers’ were assessed as scoring 14 on the matrix, indicating a relatively high risk of exposure due to characteristics inherent in their job. However, the threshold used (13+) would include half of the UK workforce and the matrix’s factors for transmission risk and mitigation could clearly mask a heterogeneity of risk as suggested in our own qualitative work [38]. This has nothing to do with the results presented here. It is a further development. I would prefer to delete it.

RESPONSE:

We included the highlighted text to show how the context for considering risk for DCWs has developed alongside our study. It was consideration of such potential risk that drove the study initially and to us it seems relevant to show how broader thinking about occupational risk has evolved. We have added the following to make this clearer:

“Differing study methodologies may reduce the value of direct comparisons, while also offering different levels of insight. Clarity of the study population is one such parameter. At the commencement of the pandemic consideration of work-place characteristics, opportunities to mitigate risk and rapidly emerging evidence of infection and associated mortality drove interest in social care and in our study, the work of DCWs.”

VERSION 3 – REVIEW

REVIEWER	Hoedl, Manuela Medical University of Graz, Institute of Nursing Science
REVIEW RETURNED	12-May-2023
GENERAL COMMENTS	None